# Spinal cord injury impairs cardiac function due to impaired bulbospinal sympathetic control

Mary P. M. Fossey[1,2,14], Shane J. T. Balthazaar [1,2,14], Jordan W. Squair[1], Alexandra M. Williams [1,3], Malihe-Sadat Poormasjedi-Meibod[1], Tom E. Nightingale[1,4,5], Erin Erskine[1,3], Brian Hayes[1], Mehdi Ahmadian [1,6], Garett S. Jackson[7], Diana V. Hunter[1], Katharine D. Currie[1], Teresa S. M. Tsang[8], Matthias Walter [1,9], Jonathan P. Little [10], Matt S. Ramer[1,11], Andrei V. Krassioukov [1,2,12,13,15✉] & Christopher R. West [1,3,15✉]

Spinal cord injury chronically alters cardiac structure and function and is associated with increased odds for cardiovascular disease. Here, we investigate the cardiac consequences of spinal cord injury on the acute-to-chronic continuum, and the contribution of altered bulbospinal sympathetic control to the decline in cardiac function following spinal cord injury. By combining experimental rat models of spinal cord injury with prospective clinical studies, we demonstrate that spinal cord injury causes a rapid and sustained reduction in left ventricular contractile function that precedes structural changes. In rodents, we experimentally demonstrate that this decline in left ventricular contractile function following spinal cord injury is underpinned by interrupted bulbospinal sympathetic control. In humans, we find that activation of the sympathetic circuitry below the level of spinal cord injury causes an immediate increase in systolic function. Our findings highlight the importance for early interventions to mitigate the cardiac functional decline following spinal cord injury.

[1] International Collaboration on Repair Discoveries (ICORD), University of British Columbia, Vancouver, BC, Canada. [2] Experimental Medicine, Department of Medicine, Faculty of Medicine, University of British Columbia, Vancouver, BC, Canada. [3] Department of Cellular and Physiological Sciences, Faculty of Medicine, University of British Columbia, Vancouver, BC, Canada. [4] School of Sport, Exercise and Rehabilitation Sciences, University of Birmingham, Birmingham, UK. [5] Centre for Trauma Sciences Research, University of Birmingham, Edgabaston, Birmingham, UK. [6] School of Kinesiology, Faculty of Education, University of British Columbia, Vancouver, BC, Canada. [7] Faculty of Health and Social Development, University of British Columbia, Kelowna, BC, Canada. [8] Division of Cardiology, University of British Columbia, Vancouver General and University of British Columbia Hospital Echocardiography Department, Vancouver, BC, Canada. [9] Department of Urology, University Hospital Basel, University of Basel, Basel, Switzerland. [10] School of Health and Exercise Sciences, University of British Columbia, Kelowna, BC, Canada. [11] Department of Zoology, Faculty of Science, University of British Columbia, Vancouver, BC, Canada. [12] Division of Physical Medicine and Rehabilitation, Faculty of Medicine, University of British Columbia, Vancouver, BC, Canada. [13] GF Strong Rehabilitation Centre, Vancouver Coastal Health, Vancouver, BC, Canada. [14] These authors contributed equally: Mary P. M. Fossey, Shane J. T. Balthazaar. [15] These authors jointly supervised this work: Andrei V. Krassioukov, Christopher R. West. ✉email: krassioukov@icord.org; chris.west@ubc.ca

Spinal cord injury (SCI) immediately interrupts connections between supraspinal structures and sympathetic preganglionic neurons in the spinal cord that innervate multiple physiological systems. When SCI is above the 6th thoracic spinal level (high-level SCI), bulbospinal input to sympathetic preganglionic neurons controlling the heart and systemic vasculature is disrupted. This disruption leads to reduced cardiac inotropic function and altered cardiac loading[1–3]. Together, altered cardiac inotropy and loading are thought to underpin the reductions in left ventricular (LV) volumes, cardiac output (Q), and estimated LV mass that are observed clinically in individuals with chronic SCI[2,4–6]. Our group has extended these clinical findings using invasive techniques in preclinical models of SCI in which we have demonstrated reductions in LV contractility (end-systolic elastance; $E_{es}$)[7,8] and cardiomyocyte atrophy[7,9], along with an associated upregulation of proteolytic pathways in LV tissue[7]. These SCI-induced alterations to cardio-autonomic function, in addition to other cardio-metabolic sequelae (e.g., alterations in physical activity[10], metabolism[11–17], hemodynamics[18,19], and arterial stiffness[20]) likely contribute to the increase in the incidence of acute cardiovascular events[21,22] and the odds for chronic cardiovascular disease post SCI[23]. Such changes also limit maximal Q during exercise[24], which may ultimately reduce the efficacy of exercise interventions to offset cardio-metabolic disease in those with high-level SCI[25]. While the cardiac consequences of SCI are becoming increasingly characterized[2,7–9], the underlying mechanisms responsible for inducing these changes as well as the temporal development from acute-to-chronic SCI remain unclear, which precludes researchers and clinicians from optimizing treatment strategies for patients with SCI.

To understand these mechanisms, we conducted a series of translational experiments across humans and rats with high-level SCI to first determine the progression of cardiac changes post SCI and validate our experimental model (Part I; Fig. 1). We subsequently use our model to identify the role that reduced bulbospinal input to sympathetic preganglionic neurons plays in cardiac functional decline post SCI (Part II; Fig. 1). In the final part, we translate our experimental mechanistic findings back to the clinical population by studying the impact that activating the sublesional sympathetic circuitry has on the heart and circulation (Part III; Fig. 1). In Part I, we report that humans with cervical SCI have reduced LV volumes and lower LV twist mechanics in the chronic phase (≥12 months post SCI),[26] but preserved cardiac function in the sub-acute phase of SCI (1–6 months post SCI)[26]. We also validate the clinical utility of our experimental rat model of SCI by demonstrating that the reduction in LV volumes and size of the cardiomyocytes is not fully present until the chronic phase post SCI, but importantly show that SCI causes an immediate and sustained reduction in LV contractile function that is akin to those observed in animal models of heart failure. In Part II, we first identify the neural origin of LV contractile dysfunction post SCI in rats by demonstrating that a chemical ganglionic blockade (hexamethonium bromide; HEX) following a complete T3 transection (T3-SCI) produces no further reduction to LV systolic function. We next find that sparing bulbospinal sympathetic fibers or preserving these pathways with a neuroprotective approach (i.e., minocycline) prevents the decline in LV contractile function seen in rats with SCI. In Part III, we show in humans with cervical SCI that activating the sympathetic circuitry below the injury via penile vibrostimulation (PVS) acutely normalizes LV function. Collectively, our findings implicate the SCI-induced loss of bulbospinal sympathetic control as a primary player in the rapid and sustained reduction in cardiac function post SCI, and suggest that interventions that target these pathways and can be applied during the acute/sub-acute setting before structural adaptations in the LV begin to occur should be further developed to improve cardiac function post SCI.

## Results

**Temporal progression of cardiac changes post SCI.** To assess the temporal changes in cardiac structure and function following high-level SCI (Part I), we prospectively recruited a large cohort of individuals ($n = 59$) with sub-acute (≤5 months post SCI) and chronic cervical SCI (≥24 months post SCI), as well as non-injured controls, and used 2D transthoracic echocardiography to measure LV structure, function, and mechanics (Fig. 2a; Supplementary Data 1 and Supplementary Table 1). We found that LV volumes (Fig. 2b–e) and systolic velocity (S′) were lower in individuals with chronic SCI vs. non-injured controls. We also found that early filling velocity (E′) was lower and the early diastolic filling to early myocardial relaxation (E/E′) ratio was higher in individuals with chronic SCI vs. non-injured controls, potentially implying altered diastolic function. With respect to LV mechanics, we found that individuals with chronic cervical SCI had significantly lower peak LV twist (Fig. 2f, i) that predominantly resulted from lowered peak apical rotation (Fig. 2h, l) vs. those with sub-acute SCI. Reduced peak LV twist was accompanied by slower peak systolic twist velocity (Fig. 2g, j) and slower untwisting rate (Fig. 2g, k) in individuals with chronic cervical SCI vs. those with sub-acute cervical SCI and non-injured controls. There were otherwise no differences between any groups for peak basal rotation, global circumferential strain, or longitudinal strain. Our clinical findings suggest that the reductions in LV volumes and mechanics following cervical SCI take months to manifest in humans, potentially identifying a therapeutic window of opportunity for interventions in the sub-acute phase.

To validate that our complete T3 transection rat model of SCI induces a similar cardiac phenotype to that observed clinically, we conducted a longitudinal study using in vivo echocardiography to track changes in LV dimensions and volumes, as well as global systolic and diastolic function over the first 8 weeks following T3-SCI or SHAM injury (Fig. 3a; Supplementary Data 2, 3). Similar to our observations in the clinical cohort, we show that rats with SCI had gradual reductions in LV volumes (i.e., end-diastolic volume (EDV) and stroke volume (SV)) and Q over time, all of which were fully manifested by 8 weeks post-injury (Fig. 3c–e).

Though echocardiography provides a useful tool for tracking changes in cardiac function over time, all of the measured metrics are considered to be load-dependent (i.e., sensitive to changes in preload and afterload), which is problematic given preload and afterload change with SCI. In this respect, in vivo LV catheterization can circumvent these limitations by providing additional load-independent metrics of cardiac systolic and diastolic function which are not sensitive to changes in LV preload and afterload. We therefore conducted a cross-sectional study in rats with T3-SCI or SHAM where we performed terminal closed-chest invasive in vivo LV catheterization to assess both peripheral hemodynamics and LV function at different time-points ranging from 1 day to 8 weeks post SCI (Fig. 4a, b; Supplementary Table 2). Systolic blood pressure (SBP) and heart rate (HR) were reduced in all SCI groups beginning 1 day post SCI (vs. SHAM). Mean arterial pressure (MAP) tended to be lower from 1-day post SCI and was significantly lowered from 5 days post SCI (Fig. 4i). With respect to LV pressures, SCI caused immediate and persistent reductions in maximum LV pressure ($P_{max}$), the maximal rate of LV pressure generation ($dP/dt_{max}$), and LV $E_{es}$, which collectively indicate impaired cardiac contractile function as early as 1-day post SCI (Fig. 4c, e–h). SCI also impaired the maximal rate of diastolic pressure decay ($-dP/dt_{min}$), but did not influence end-diastolic pressure ($P_{ed}$) or the diastolic time constant (tau). Ventricular-vascular coupling ($E_a/E_{es}$), an index of coupling efficiency between the heart and the arterial system, was augmented (uncoupled) as early as 1-day post SCI. Together our temporal in vivo preclinical findings suggest

that high-level SCI causes an immediate reduction in LV contractile function and an uncoupling of the heart and blood vessels, which occurs prior to changes in LV volumes.

To estimate the temporal changes in sympathetic activity post SCI, we measured plasma norepinephrine (NE) at different

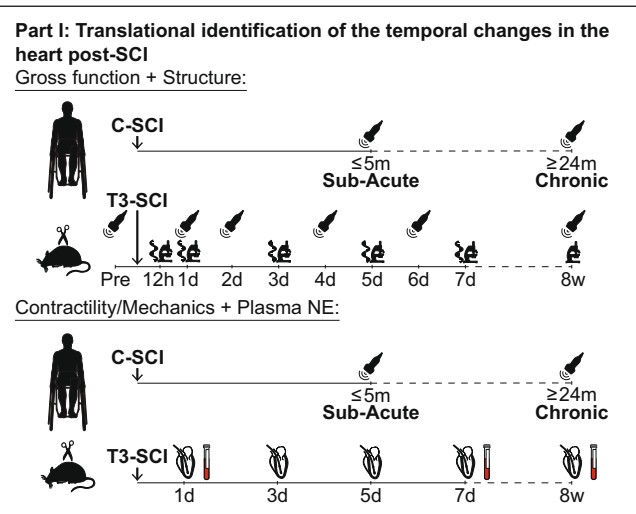

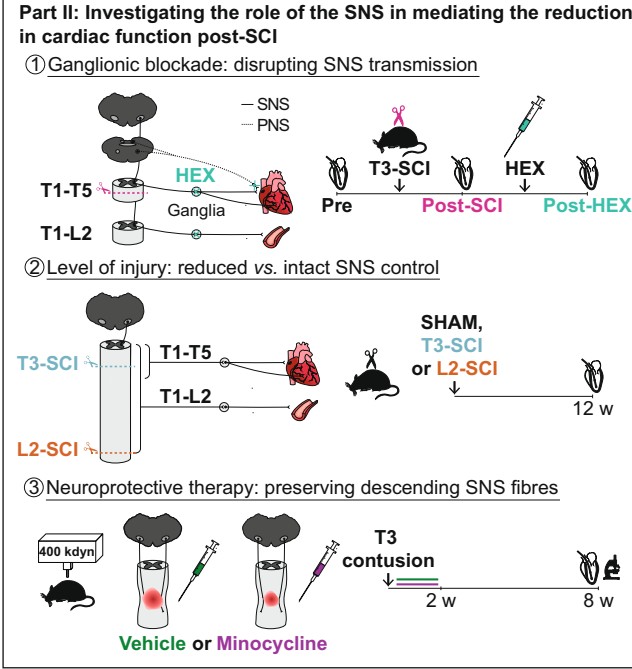

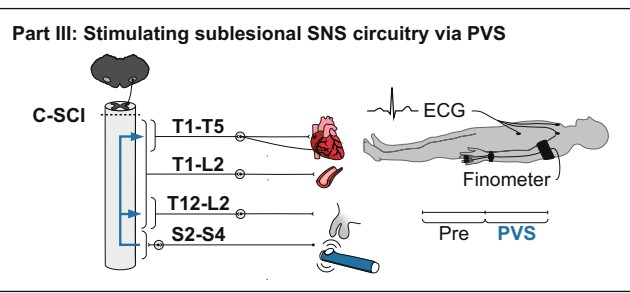

**Fig. 1 Summary of aims and methods.** C-SCI, cervical spinal cord injury; ECG, electrocardiogram; HEX, hexamethonium bromide; L2-SCI, L2 complete transection SCI; PNS, parasympathetic nervous system; PVS, penile vibrostimulation; SNS, sympathetic nervous system; T3-SCI, T3 complete transection SCI.

timepoints post SCI using an enzyme-linked immunosorbent assay (ELISA) (Fig. 4d; Supplementary Table 2). We found that circulatory plasma NE was lower as early as 1-day post SCI and remained depressed below SHAM levels at 8 weeks post SCI.

To establish the timeline of cardiac structural remodeling in response to SCI, we investigated changes in the expression of genes known to be involved in cardiac proteolysis and additionally measured cardiomyocyte length, width, and cross-sectional area at each of our acute and chronic timepoints (Fig. 5a; Supplementary Tables 3, 4). We performed RNA extraction and quantitative real-time polymerase chain reaction (qPCR) to investigate gene expression of key markers of protein degradation pathways involved in the ubiquitin-proteasome system (UPS) and autophagy pathways in tissue extracted from the LV free wall. The RNA fold change of UPS marker atrogin-1 (*MAFbx*) was elevated 12 h post SCI by approximately 3-fold and remained elevated at 7 days post SCI (Fig. 5b). SCI did not induce changes in the gene expression of UPS marker muscle RING-finger protein (*MuRF1*; Fig. 5c) and autophagy markers (Fig. 5d). To assess gross cardiomyocyte morphology, we conducted quadruple immunofluorescence staining on mid-ventricular cross-sections of the LV free wall (Fig. 5e). We found reductions in standardized cardiomyocyte length, but not width, were present from 7 days post SCI and persisted into the chronic period (Fig. 5f, g). Standardized cardiomyocyte cross-sectional area (CSA) tended to be reduced at 7 days post SCI and was significantly reduced at 8 weeks post SCI (Fig. 5h). Together, these experiments suggest that while T3-SCI causes an immediate upregulation in the transcription of UPS-related genes in LV tissue, SCI-induced cardiomyocyte atrophy occurs subsequent to changes in cardiac function.

**Loss of bulbospinal sympathetic control reduces LV function.** Our findings in the temporal study (Part I) demonstrated that LV contractile function is impaired within 1-day post SCI and precedes the reduction in gross structure of the LV and the cardiomyocytes. We therefore leveraged our validated rodent model of SCI to experimentally determine whether such reductions in contractility were mediated by a loss of bulbospinal sympathetic control, and if so, whether preserving these pathways resulted in improved cardiac function.

We first sought to demonstrate that the changes in cardiac function following SCI are neurally mediated by performing a chemical ganglionic blockade post SCI. To disrupt both sympathetic and parasympathetic control, we used I.V. HEX, as it blocks nicotinic acetylcholine receptors located in the ganglia[27–29]. Utilizing an acute preparation, the rat was placed under urethane anesthesia, intubated, and underwent LV catheterization to enable LV measurements pre-intervention, post SCI, and post SCI + HEX (Fig. 6a, b; Supplementary Table 5). There were significant reductions in LV pressure-generating capacity post SCI and post SCI + HEX vs. pre-intervention, however, there were no differences in absolute cardiac indices between post SCI vs. post SCI + HEX (Fig. 6c–g).

Since we found that the decline in LV pressure-generating capacity following high-level SCI was primarily neurally mediated, we next investigated the specific involvement of bulbospinal sympathetic control in mediating this decline. To do this, rats were randomized to receive either a T3-SCI, which almost completely removes bulbospinal sympathetic control to the heart, a L2 complete transection SCI (L2-SCI), which preserves bulbospinal sympathetic control, or SHAM injury (Fig. 7a, b; Supplementary Table 6). Rats survived until 12 weeks post-injury to ensure we could fully characterize the chronic adaptations induced by these two injury paradigms. At 12 weeks

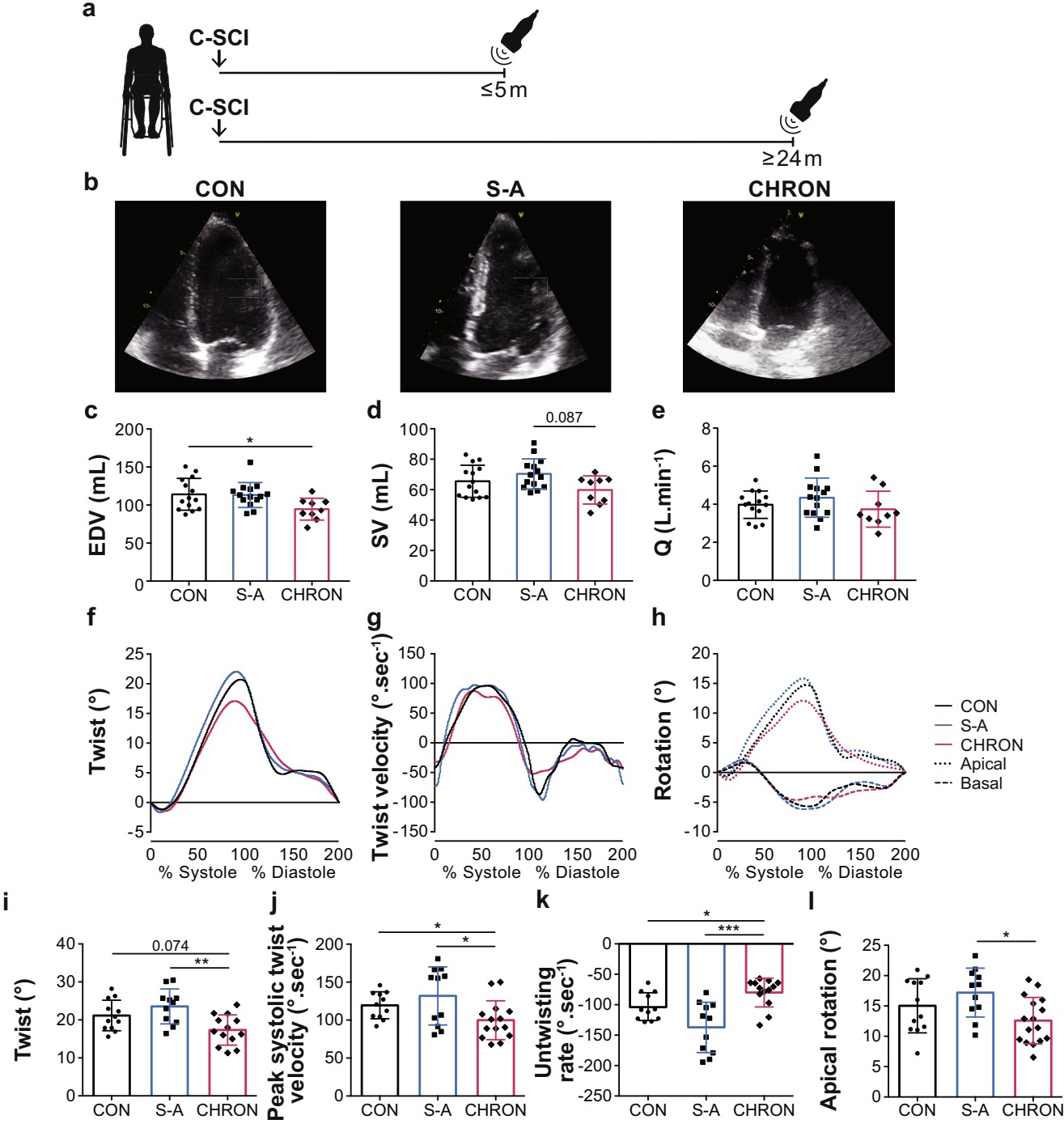

**Fig. 2 Temporal progression of echocardiography-derived LV functional, volumetric, and mechanical outcomes in human participants with cervical SCI.**
**a** Echocardiography was performed cross-sectionally on non-injured control individuals (CON), and on cervical SCI individuals at ≤5 months in the sub-acute period (S-A) and at ≥24 months in the chronic period post SCI (CHRON). **b** Representative apical four-chamber images of previously described groups (scaled to each other). **c** End-diastolic volume (EDV) was reduced in CHRON vs. CON with similar trends for **d** stroke volume (SV) and **e** cardiac output (Q). Panels **c–e** represent data from $n = 14$ CON, $n = 14$ S-A, $n = 9$ CHRON). **f** Ensemble-averaged twist, **g** twist velocity, and **h** rotation curves during one cardiac cycle across all participants in each group (panels represent data from $n = 11/12$ CON, 11 S-A, and $n = 13–16$ CHRON). **i** Twist was lower in CHRON vs. S-A and tended to be lower in CHRON vs. CON. **j** Peak systolic twist velocity was slower in CHRON vs. both CON and S-A. **k** Untwisting rate was slower in CHRON vs. both CON and S-A. **l** Apical rotation was lower in CHRON vs. S-A, while basal rotation was not. Bar graph data are presented as mean ± SD with individuals represented as symbols ($n = 9–16$/group) and were analyzed using a one-way ANOVA and Tukey HSD post hoc test: $*p < 0.05$, $**p < 0.01$, and $***p < 0.001$. See Supplementary data 1 for detailed statistics and other echocardiography-derived outcomes. See Supplementary Table 1 for participant characteristics. Source data are provided for all variables shown in this figure in the source data file.

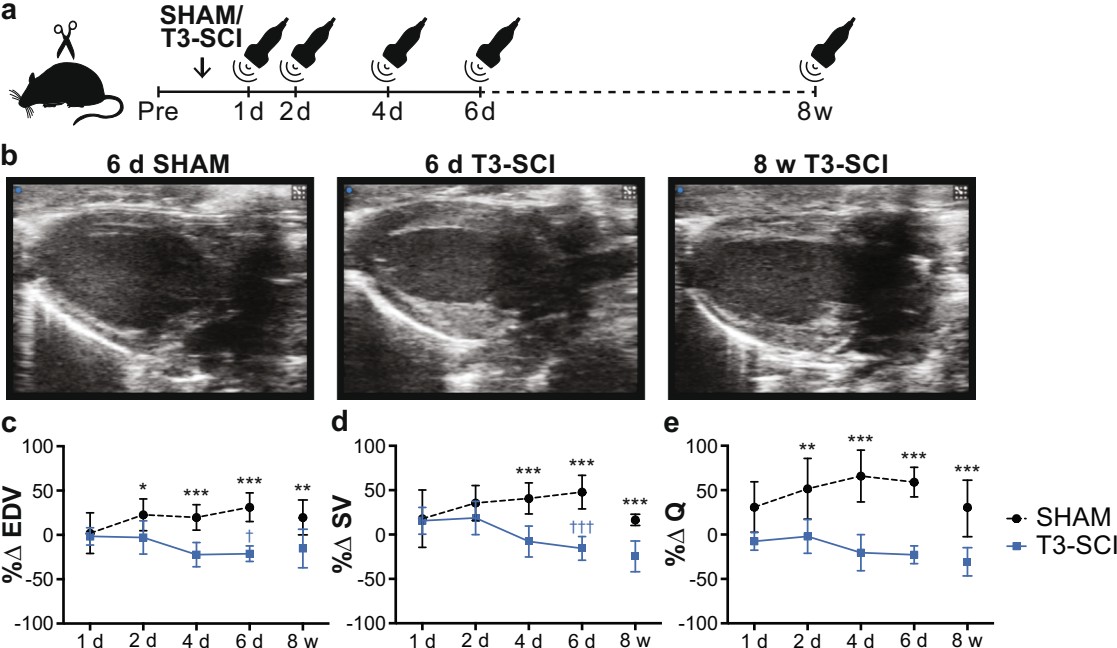

**Fig. 3 Temporal progression of echocardiography-derived functional and volumetric outcomes in rats with T3 complete transection SCI (T3-SCI).**
**a** Male Wistar rats (10–11 wks of age) underwent either SHAM or T3-SCI. Echocardiography was performed longitudinally in an acute group at pre-surgery, 1 day, 2 days, 4 days, and 6 days post SCI/SHAM surgery, and in a chronic group at pre-SCI/SHAM and at 8 weeks post SCI/SHAM to ensure appropriate age-matching (under isoflurane anesthesia). **b** Representative parasternal long-axis images of SHAM (6 days post-SHAM) and T3-SCI rats (6 days and 8 weeks post SCI) at end-diastole at identical scaling. **c** Percent change (%Δ) in EDV ($n = 6$–8/group), **d** SV ($n = 6$–8/group), and **e** cardiac output (Q; $n = 6$–8/group) were significantly reduced acutely and chronically post T3-SCI vs. post-SHAM and compared to day 1 post-injury. Data are presented as mean ± SD. Acute %Δ data ($n = 6$–8/group) were analyzed using a two-way repeated-measures ANOVA (factors: time and group) with Bonferroni corrected post hoc pairwise comparisons (between and within groups). Chronic %Δ data ($n = 6$–10/group) was analyzed using a two-sample $t$-test. Between-group comparison: *$p < 0.05$, **$p < 0.01$ and ***$p < 0.001$. Within-group comparison to 1-day post SCI/SHAM: †$p < 0.05$ and †††$p < 0.001$. See Supplementary Data 2, 3 for detailed statistics. Source data are provided for all variables shown in this figure in the source data file.

post SCI, rats underwent the same terminal invasive in vivo LV catheterization preparation described above. Peripheral hemodynamics (e.g., MAP, SBP, and DBP; Fig. 7h) as well as indices of LV pressure-generating capacity (Fig. 7c, f, g) and contractility (Fig. 7d, e) were lower following T3-SCI vs. SHAM and L2-SCI, and ventricular-vascular coupling ($E_a/E_{es}$) was uncoupled following T3-SCI vs. SHAM and L2-SCI. In contrast, when the bulbospinal pathways were intact (L2-SCI) we did not observe any significant difference in LV function compared to SHAM animals.

It is possible that by inducing SCIs at different levels in the spinal cord (i.e., T3 vs. L2-SCI), other indices which may influence cardiac function, such as in cage activity, circulating hormones, and the degree of plasticity within the sympathetic nervous system (SNS) may differ between groups and impact our findings. We therefore devised an additional experimental paradigm in an effort to specifically isolate the influence of the bulbospinal sympathetic pathways on the heart within the T3-SCI model. To do so, we leveraged the neuroprotective effects of minocycline, a prototypical neuroprotective agent for the field which we have previously reported to be effective in preserving approximately 10–15% of descending spinal sympathetic pathways following T3 contusion SCI[30]. Animals were randomized to receive either minocycline or vehicle every 12 h for 2 weeks post SCI and were assessed for LV function 8 weeks post SCI (Fig. 8a, b; Supplementary Table 7). Retrograde tracing of the bulbospinal sympathetic axons via FluoroGold injections at the T8 spinal level confirmed our previous observation that minocycline spares a greater number of descending bulbospinal sympathetic pathways compared to vehicle (Fig. 8h, i)[30]. From our terminal

cardiovascular assessment, we found that LV contractility (Fig. 8c, d) and pressure-generating capacity (Fig. 8e, f) were significantly higher in minocycline-treated vs. vehicle-treated rats, while there was no difference in MAP between the two groups (Fig. 8g).

Collectively, the preclinical findings from experiments in Part II imply the reduction in LV pressure-generating capacity and contractility post SCI are neurally mediated and result from the loss of bulbospinal sympathetic input to the sympathetic preganglionic neurons, that ultimately control the heart. Preservation of just 10–15% of these pathways[30] is sufficient to normalize LV function, implying that the heart may be a conducive target for testing neuroprotective treatment efficacy in the field of SCI.

**Activating sublesional SNS circuitry improves LV function.** In the chronic setting post SCI, there is good evidence that the sublesional sympathetic circuitry between the spinal cord and the periphery remains intact and conducive to activation. For instance, both epidural and transcutaneous stimulation of the lower thoracic spinal cord improve (increase) blood pressure post SCI[31,32]. We therefore sought to determine whether activating the sublesional spinal sympathetic circuitry is able to acutely restore cardiac function in the chronic stage post SCI (Part III). We analyzed beat-by-beat hemodynamics (i.e., blood pressure and HR) via finometry and electrocardiogram (ECG) monitoring, and estimated LV systolic function (i.e., dP/dt$_{max}$) via model flow during PVS in ten individuals with chronic cervical SCI (Fig. 9b; Supplementary Table 8). PVS stimulates the sublesional sympathetic preganglionic neurons via the activation of peripheral

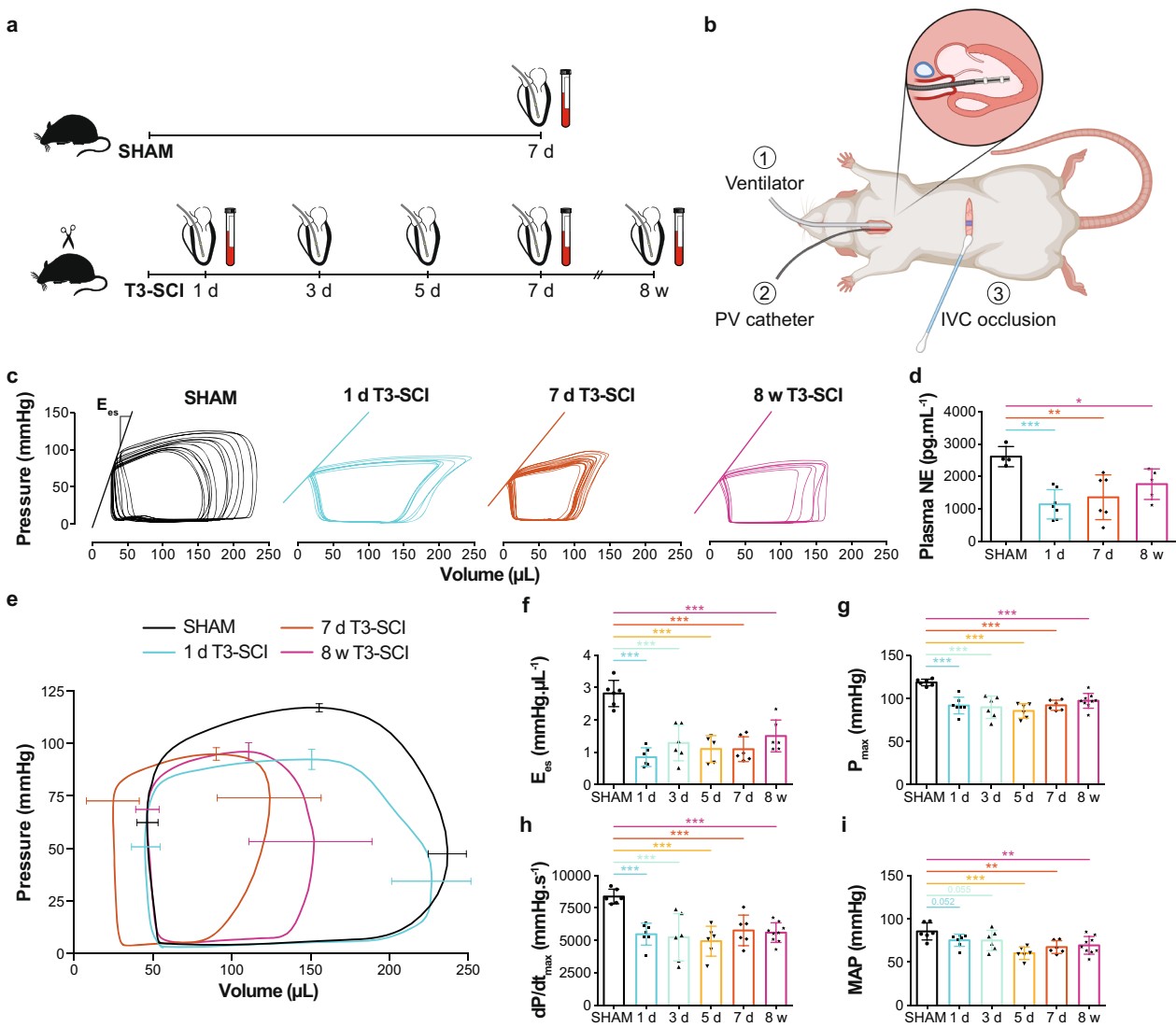

**Fig. 4 Temporal progression of changes in catheter-derived hemodynamics, LV function, and plasma norepinephrine (NE) in rats with SHAM injury or T3 complete transection SCI (T3-SCI). a** Methods timeline (cross-sectional): T3-SCI rats underwent pressure-volume (PV) catheterization (terminal procedure) at multiple timepoints across the acute-to-chronic continuum post SCI (1 day, 3 days, 5 days, 7 days, and 8 weeks). SHAM rats were catheterized at 7 days post-SHAM. The PV catheter was placed in the right carotid artery to assess arterial hemodynamics, and then was advanced into the LV to assess load-dependent LV function. To assess load-independent LV function, in particular end-systolic elastance ($E_{es}$, index of contractility), the inferior vena cava (IVC) was occluded with a cotton-tip applicator for transient reductions to preload. To assess sympathetic activity, plasma NE was measured via ELISA (7 days post-SHAM; 1 day, 7 days, and 8 weeks post SCI). **b** Surgical setup at termination illustrating: (1) the connection to a ventilator, performed via tracheotomy and intubation, (2) the PV catheter inserted into the LV via the right carotid artery, and (3) the abdominal incision for IVC occlusions (created with BioRender.com). **c** Representative IVC occlusions with lines representing $E_{es}$ for SHAM, 1 day, 7 days, and 8 weeks T3-SCI (other groups in Supplementary Table 2). **d** Plasma NE was reduced as soon as 1-day post SCI vs. SHAM ($n = 4$–7/group). **e** Averaged PV loops with SD bars for SHAM, 1 day, 7 days, and 8 weeks T3-SCI (other groups in Supplementary Table 2). **f** $E_{es}$ ($n = 6$/group) and pressure indices such as **g** maximum pressure ($P_{max}$; $n = 6$–9/group) and **h** maximal rate of systolic pressure increment (dP/dt$_{max}$; $n = 6$–9/group) were rapidly reduced starting at 1-day post SCI vs. SHAM. **i** Mean arterial pressure (MAP; $n = 6$–9/group) was reduced starting at 5 days post SCI vs. SHAM. Data are presented as mean ± SD with individual rats represented as symbols, and were analyzed using a linear mixed regression model with coefficient (group mean differences) $t$-test $p$-values vs. SHAM: *$p < 0.05$, **$p < 0.01$, and ***$p < 0.001$. See Supplementary Table 2 for detailed statistics and other functional outcomes. Source data are provided for all variables shown in this figure in the source data file.

afferents and interneurons[33]. PVS significantly increased MAP, peak model-flow derived SV and dP/dt$_{max}$ (Fig. 9a, c, e, f), and reduced peak HR (Fig. 9a, d).

## Discussion

Our findings demonstrate that high-level SCI causes a rapid and sustained reduction in LV contractile function that precedes

structural adaptations at the cellular and organ levels, and occurs due to the SCI-induced loss of bulbospinal sympathetic control. Preserving bulbospinal sympathetic circuitry in the acute setting post SCI or activating the sublesional sympathetic circuitry in the chronic setting post SCI both improve cardiac function.

In a large cohort of individuals with cervical SCI we demonstrate that chronic, but not sub-acute, SCI is associated with a reduction in LV volumes and mechanics. In our rat model of SCI,

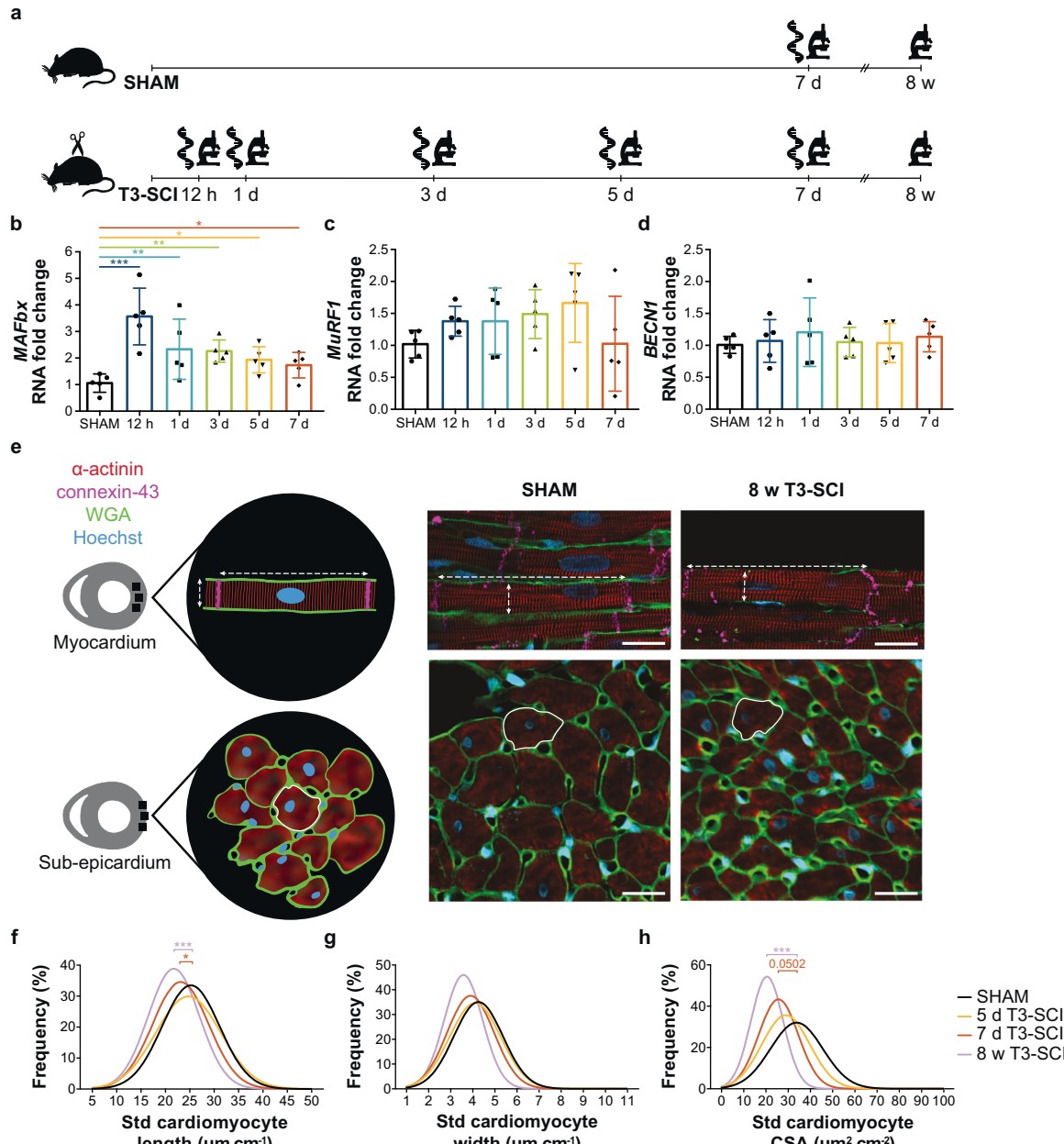

**Fig. 5 Temporal progression of changes in gene expression of proteolytic pathways in LV tissue and cardiomyocyte structure from rats with SHAM injury or T3 complete transection SCI (T3-SCI). a** Methods timeline (cross-sectional): LV tissue was collected from T3-SCI rats across the acute-to-chronic continuum post SCI to assess proteolytic gene expression (qPCR; 12 h, 1 day, 3 days, 5 days, and 7 days) and cardiomyocyte dimensions (immunofluorescence; 1 day, 3 days, 5 days, 7 days, and 8 weeks). SHAM tissue was collected at 7 days post-SHAM. **b** The RNA fold change of ubiquitin-proteasome system (UPS) marker atrogin-1 (*MAFbx*) was elevated at all acute timepoints post SCI vs. SHAM, while UPS marker **c** muscle RING-finger protein (*MuRF1*) and **d** autophagy marker beclin-1 (*BECN1*) showed no changes in gene expression. Gene expression data are presented as mean ± SD with individual rats represented as symbols (*n* = 5/group). **e** On the left is a schematic representation of where cardiomyocytes were imaged with methods of measurement. On the right, representative immunofluorescence images of longitudinally oriented cardiomyocytes located in the myocardium of the LV free wall shown in the top row with length and width represented as dotted white lines (wheat germ agglutinin, WGA). Cross-sectionally oriented cardiomyocytes located in the sub-epicardium of the LV free wall are shown in the bottom row with cross-sectional area (CSA) contoured with a solid white line. Scale, 20 mm. For replication, a minimum of 130 lengths, 210 widths, and 98 CSAs were averaged per animal. **f** Frequency distributions as Gaussian curves centered at the mean of standardized to femur length (std) cardiomyocyte length, **g** width (*n* = 5–9/group), and **h** CSA (*n* = 5–9/group) are shown; only the 5 and 7 days T3-SCI groups are represented for the acute timepoints for clarity. Std length was reduced at 7 days and 8 weeks post SCI vs. SHAM (left shifts), while std CSA tended to be reduced at 7 days post SCI and was reduced at 8 weeks post SCI vs. SHAM. Gene expression and cardiomyocyte measures were analyzed using a linear mixed regression model vs. SHAM with coefficient (group mean differences) *t*-test *p*-values: *$p < 0.05$, **$p < 0.01$, and ***$p < 0.001$. See Supplementary Table 3 for all histological and molecular outcomes. Source data are provided for all variables shown in this figure in the source data file.

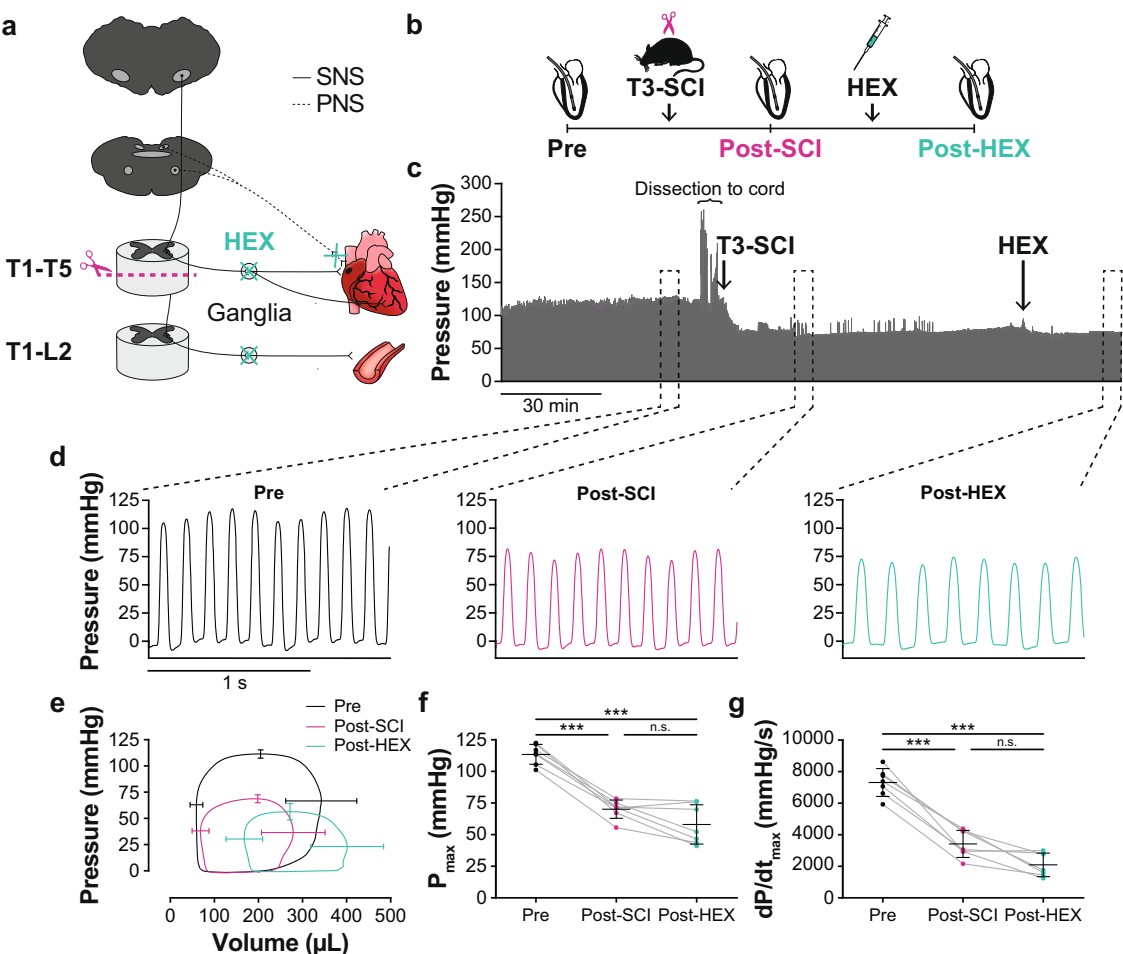

**Fig. 6 Effects of a T3 complete transection SCI (T3-SCI) and ganglionic blockade with hexamethonium (HEX) on catheter-derived LV functional outcomes in rats. a** Neuroanatomical representation of study design and interventions. **b** Methods timeline (longitudinal): Male Wistar rats (10–11 wks of age) underwent first LV catheterization to record pressure-volume data, second a T3-SCI, and last an intravenous HEX infusion (20 mg/kg). **c** A representative LV pressure trace of the entire data recording is shown on top (~3 h). One minute of data was analyzed at pre-intervention, nadir post SCI, and once stabilized post-HEX. The three zoomed-in traces show beat-by-beat pressure at pre-intervention, post SCI, and post-HEX (from left to right). **d** Example raw pressure waveforms under each blockade/condition. **e** Ensemble-averaged pressure-volume loops of each stage are shown with SD bars. **f** Maximum pressure ($P_{max}$) and **g** maximal rate of LV pressure generation ($dP/dt_{max}$) reduced following post SCI compared to pre-intervention, and did not further reduce following a HEX infusion post SCI. Data are presented as mean ± SD with individual rats represented as symbols ($n = 7$) and were analyzed using a one-way repeated-measures ANOVA with pairwise dependent samples $t$-test post hoc, Bonferroni corrected: ***$p < 0.001$; n.s., non-significant. See Supplementary Table 5 for detailed statistics and other catheter-derived outcomes. Source data are provided for all variables shown in this figure in the source data file.

we replicate these findings by demonstrating gradual reductions in LV volumes and cardiomyocyte structure that are present as soon as 1 week post SCI and fully manifested by 8 weeks post SCI. Though fraught with challenges, extrapolative methods suggest that 1 week post SCI in rats is equivalent to ~6 months in humans[34,35] (i.e., transition between sub-acute and chronic phase), suggesting that these timelines for cardiac remodeling broadly align across species. Unlike LV volumes which take days for rats and months for humans to adapt post SCI, our rat studies demonstrate that the reductions in LV pressure-generating capacity and contractility (i.e., systolic function) occur immediately following SCI, as reflected by rapid and sustained reductions in LV $P_{max}$, $dP/dt_{max}$, and $E_{es}$. Reductions in pressure-generating capacity and contractility occurred in concert with a reduction in circulatory plasma NE, implying reduced sympathetic tone at these timepoints (see also Part II below). The temporal differences between the immediate changes in cardiac pressure generation vs. delayed reductions in LV volumes (i.e., EDV and SV) and cardiomyocyte morphology suggest that their underlying stimuli

may differ. Specifically, it seems very likely that pressure-derived indices of LV systolic function are much more dependent on the sole influence of the SNS, which is responsible for maintaining vasomotor tone and cardiac inotropy (discussed in Part II below)[36]. Conversely, the more gradual nature of the reduction in cardiac volumes suggests that factors other than interrupted bulbospinal sympathetic control contribute to the reduction in LV volumes post SCI. Outside the field of SCI, classic studies of imposed bed-rest report reduced cardiac volumes due to lower cardiac filling via the Frank-Starling relationship[37,38], that closely track reductions in plasma volume[38,39]. Although the time-course of changes in plasma volume post SCI has not been explored a number of studies have reported that blood volume is reduced in the chronic period[1,3], which is when we find evidence of reduced LV volumes.

Our molecular and histological findings demonstrate that high-level SCI causes a rapid and sustained upregulation of proteolytic pathways (UPS) which ultimately led to cardiomyocyte atrophy. In the field of skeletal muscle wasting, reduced mechanical

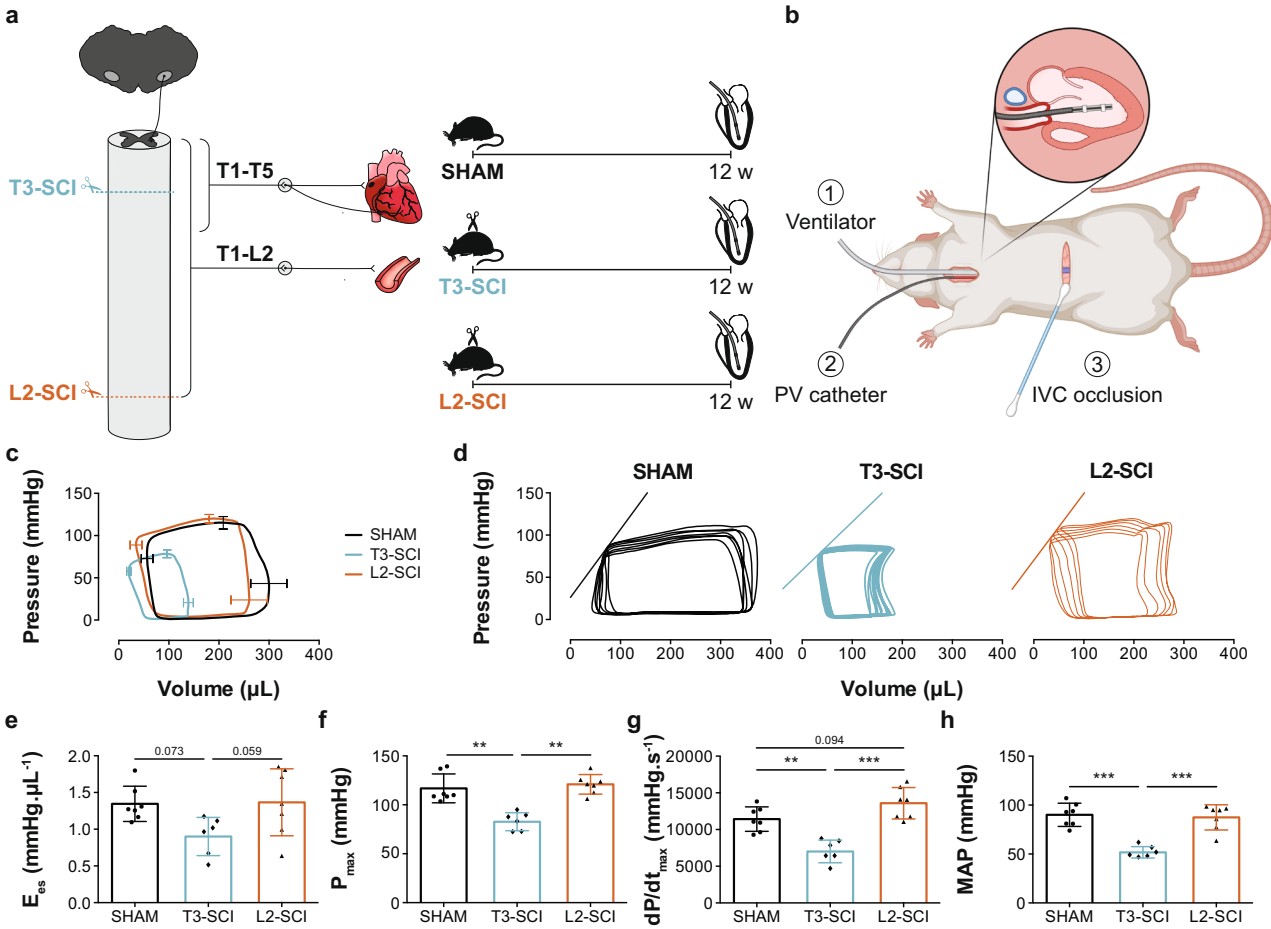

**Fig. 7 Effects of level of injury on catheter-derived cardiovascular outcomes in rats with complete transection SCI at the T3 and L2 level (reduced vs. intact bulbospinal sympathetic control). a** Methods timeline: Male Wistar rats (10–11 wks of age) underwent SHAM injury, T3 complete transection SCI (T3-SCI) or L2 complete transection SCI (L2-SCI). Twelve weeks post-injury, rats underwent catheterization (terminal procedure) to assess hemodynamics and LV function (see Fig. 4 for catheterization details). **b** Surgical setup at termination, described in Fig. 4 and created with BioRender.com. **c** Ensemble-averaged pressure-volume loops with bars representing SD. **d** Representative inferior vena cava (IVC) occlusions with the line representing end-systolic elastance ($E_{es}$; index for contractility). **e** $E_{es}$ ($n = 6$–7/group) tended to be lower post T3-SCI vs. SHAM or L2-SCI. **f** Maximum pressure ($P_{max}$; $n = 6$–7/group), **g** maximal rate of systolic pressure increment (dP/dt$_{max}$; $n = 6$–7/group) and **h** mean arterial pressure (MAP; $n = 6$–7/group) were lower post T3-SCI vs. SHAM or L2-SCI. Data are presented as mean ± SD with individual rats represented as symbols and were analyzed using a one-way ANOVA with Tukey HSD post hoc: **$p < 0.01$ and ***$p < 0.001$. See Supplementary Table 6 for detailed statistics and other catheter-derived outcomes. Source data are provided for all variables shown in this figure in the source data file.

loading[40,41] and elevated angiotensin II (ANGII) levels[42–45] are associated with increased UPS activity. Though changes in ANGII levels[7] and cardiac mechanical unloading (e.g., changes in cardiac volumes) occur post SCI, they do so well after 12 h post SCI which is the first timepoint we observed UPS upregulation. We reason, therefore, that the loss of sympathetic 'stimuli' is likely to be the primary driver of UPS pathway activation. Evidence supporting this idea is provided by studies investigating cardiac muscle atrophy following cardiac denervation in a murine model, where associations between cardiomyocyte atrophy and reduced density of local sympathetic neurons and number of neurocardiac junctions are reported[46,47]. In addition, the expression levels of the *MAFbx* gene at 7 days post SCI in the current study were comparable to reported values by our research team 12 weeks following the same injury[7], which could suggest that *MAFbx* expression peaks at 12 h followed by a gradual reduction to a new elevated level of expression that is sustained into the chronic phase. With respect to autophagy, while none of our markers underwent transcriptional changes acutely post SCI our team has reported autophagy upregulation chronically post SCI[7]. Thus, it may be that autophagy-mediated regulation of

cardiomyocyte size takes longer to manifest, a possibility that is supported by Zaglia et al. who found autophagy activation required at least 30 days to manifest post-cardiac denervation[46]. Despite UPS upregulation hours post SCI, cardiomyocyte atrophy was not observed until 7 days post SCI in the current study, which is likely explained by insufficient time for protein degradation pathways to have measurable effects on gross cellular dimensions.

In Part II, we provide three complimentary lines of evidence that the reductions to LV pressure-generating capacity and contractility following high-level SCI were primarily mediated by the loss of bulbospinal control over the sympathetic preganglionic neurons. First, we show that the administration of HEX following SCI does not reduce LV pressure generation to a greater degree than SCI alone. Interestingly, there was still a small drop in dP/dt$_{max}$ after the delivery of HEX suggesting perhaps that the T1 spinal segment (which stays intact in our model) plays a minor role in cardiac contractility, as has been implied by anatomical studies that have reported a smaller number of SPNs located at the T1 segment as compared to other upper-thoracic spinal segments[48]. Second, by exposing rats to SCIs that disrupted

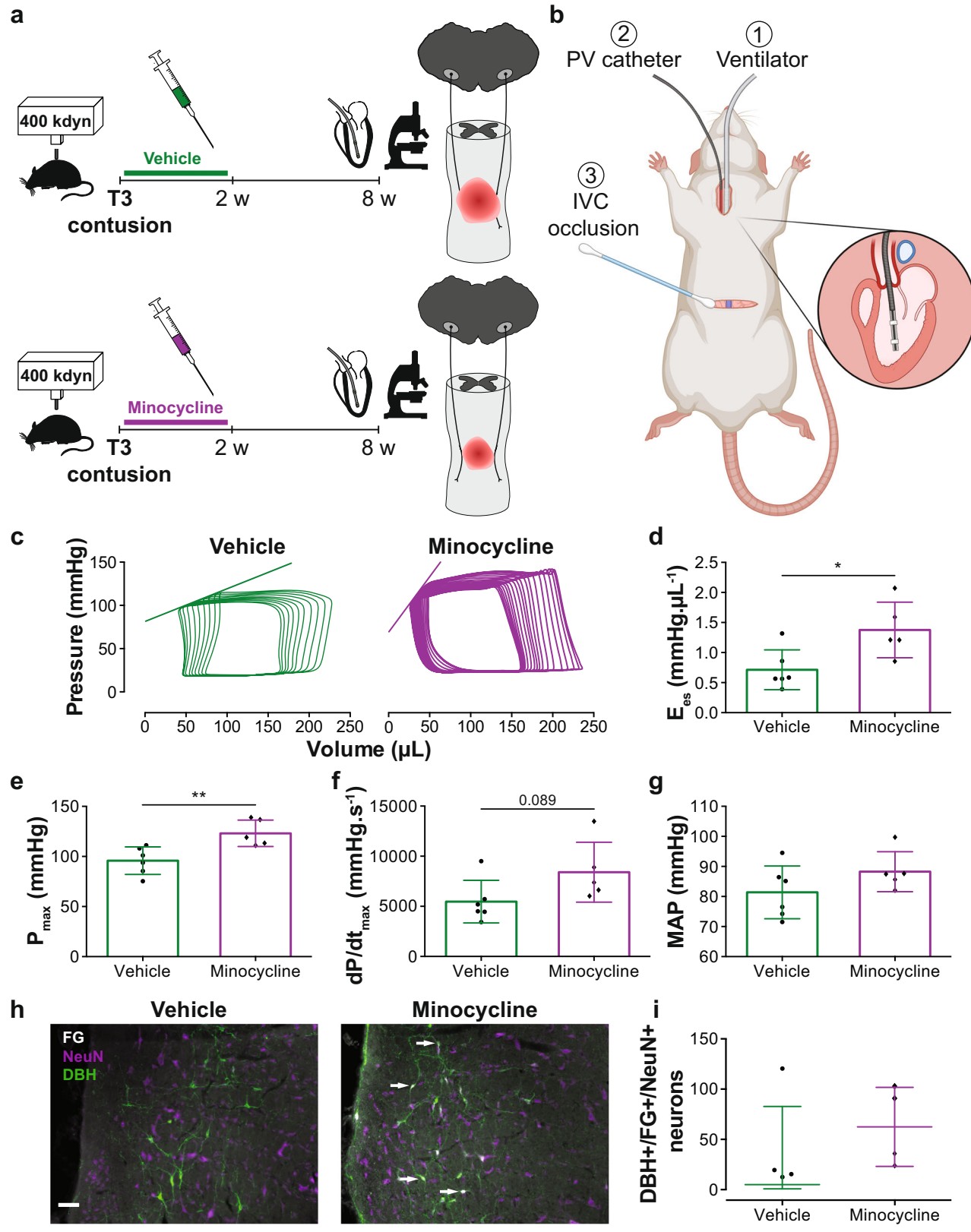

or left intact bulbospinal sympathetic projections, we found that only those rats exposed to a high-level SCI (i.e., disrupted bulbospinal sympathetic projections) exhibited impaired LV systolic function. Finally and arguably most tellingly, our minocycline experiment revealed that preserving just 10–15% of the bulbospinal sympathetic pathways[30] was sufficient to normalize LV

function in the minocycline group of this present study, despite there being no group differences in blood pressure. Collectively, these series of experiments strongly support the notion that the bulbospinal sympathetic pathways are the key regulator of LV systolic function post SCI. These observations extend our group's findings that beta-adrenergic stimulation improves LV contractile

**Fig. 8 Effects of minocycline on catheter-derived cardiovascular outcomes in rats with T3 severe contusion SCI. a** Methods timeline: T3 contused (400 kdyn, 5 s dwell time) male Wistar rats (10–11 wks of age) were administered (I.P.) vehicle (control) or minocycline every 12 h for 2 weeks. At 8 weeks post SCI, catheterization (terminal procedure) was performed to assess hemodynamics and left ventricular (LV) function (see Fig. 4 for catheterization details). **b** Surgical setup at termination, described in Fig. 4 and created with BioRender.com. **c** Representative inferior vena cava (IVC) occlusions with the line representing end-systolic elastance ($E_{es}$; index of contractility). **d** $E_{es}$ ($n = 5$–$6$/group), **e** maximum pressure ($P_{max}$; $n = 5$–$6$/group), and **f** maximal rate of systolic pressure increment ($dP/dt_{max}$; $n = 5$–$6$/group) were significantly higher in minocycline-treated vs. vehicle-treated rats. **g** Mean arterial pressure (MAP) was not significantly different between the groups. **h** Representative immunofluorescence images of the rostral ventrolateral medulla with dopamine beta-hydroxylase (DBH; green), FluoroGold (FG; white), and neuronal nuclei (NeuN; purple) stains. Arrows point to DBH/FG/NeuN positive cells. For replication please note that the count represents the number of FluoroGold/DBH-positive neurons located in the RVLM. Scale bar $= 100$ μm. **i** Minocycline-treated rats have more DBH/FG/NeuN positive cells than vehicle ($n = 4$/group). Catheter-derived data are presented as mean ± SD, and histological data are presented as median with the interquartile range. Individual rats are represented as symbols. All data in panels **d**–**g** were analyzed using a two-way independent $t$-test: *$p < 0.05$ and **$p < 0.01$. See Supplementary Table 7 for detailed statistics and other catheter-derived outcomes. Source data are provided for all variables shown in this figure in the source data file.

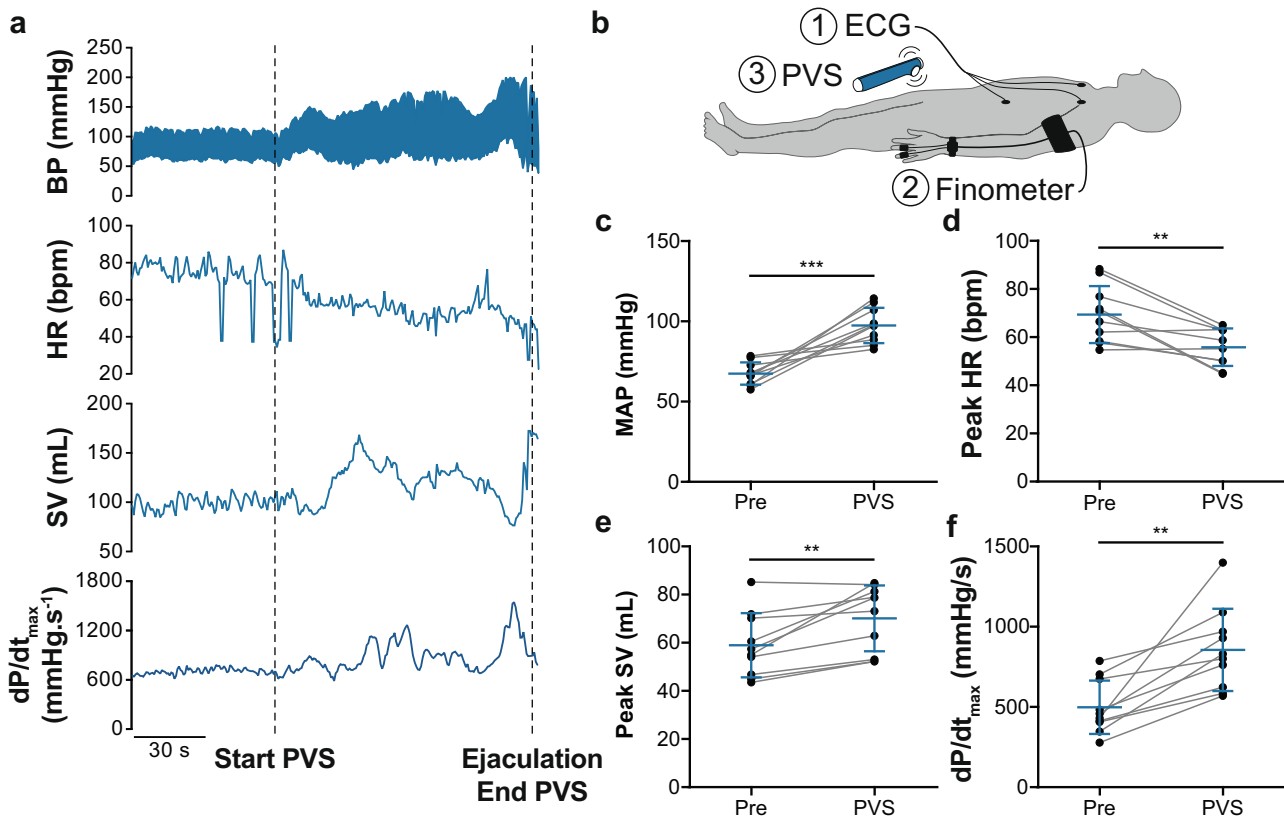

**Fig. 9 Effects of penile vibrostimulation (PVS) on model-flow-derived cardiovascular outcomes in three individuals with chronic cervical SCI. a** Representative data from one individual with a chronic cervical SCI undergoing PVS for sperm retrieval (complete C7 SCI; 41 years old; 13 years since SCI; AIS A). Note that upon the start of the stimulation with personal vibrator, there is a gradual increase in blood pressure (BP) and a reduction in heart rate (HR) that classically occurs in response to activation of the sympathetic nervous system during autonomic dysreflexia. Note that both stroke volume (SV) and left ventricular (LV) contractility ($dP/dt_{max}$) also increased during PVS with values peaking at the time of maximal activation (ejaculation). **b** Cardiovascular measures were recorded via (1) electrocardiogram (ECG) and (2) finometry, while undergoing (3) PVS. **c** Mean arterial blood pressure (MAP), **d** peak HR, **e** peak SV, and **f** $dP/dt_{max}$ were obtained prior to and during PVS, demonstrating a consistent increase in cardiac function in response to activation of the sympathetic nervous system via PVS. Data are presented as mean ± SD with individuals represented as symbols ($n = 10$) and were analyzed using paired $t$-tests: **$p < 0.01$ and ***$p < 0.001$. See Supplementary Table 8 for detailed statistics and other finometry-derived outcomes. Source data are provided for all variables shown in this figure in the source data file.

function post SCI[8], and compliment reports from studies examining heart-transplant patients (i.e., improved cardiac response to exercise post-sympathetic reinnervation)[49] and sympathectomy (i.e., reduced LV contractile function post-chemical sympathectomy in rats)[46,50,51], by highlighting the importance that the SNS plays in mediating LV contractile function post SCI. This is an important step forward for the field as it likely explains why

individuals with high-level SCI fail to mount an appropriate cardio-excitatory response to exercise[52,53], and are more resistant to cardiovascular adaptations to exercise interventions[54]. Although our current study has focused primarily on the rostral ventrolateral medulla (RVLM) as the major source of reduced sympathetic input post SCI, there are likely several additional brainstem regions that significantly contribute to regulating

sympathetic outflow and cardiovascular function post SCI, including the paraventricular nucleus of the hypothalamus[55] and the Raphe nuclei[56].

The extreme example of sublesional sympathetic circuitry activation in our PVS study demonstrate the potential to offset the reduction in cardiac function via sympathetic activation. PVS is known to elicit autonomic dysreflexia (AD), which describes a reflex arc initiated by visceral afferents that ultimately results in profound sympatho-excitation[19]. While the individuals in our study had a cardio-excitatory response to PVS, the initiation of AD can be life-threatening and should therefore only be performed under carefully controlled clinical settings. Nonetheless, our PVS findings do provide compelling support for future studies to focus on activating sublesional sympathetic circuitry in a controlled fashion as a way to offset reductions in cardiac function. Indeed, recent empirical evidence that supports a cardio-beneficial effect of activating the sympathetic circuitry is the finding that both transcutaneous and epidural electrical stimulation of the spinal cord has led to increases in cardiovascular function via the excitement of dorsal roots, which activates intraspinal circuitry to depolarize the sympathetic preganglionic neurons[57,58]. This research, therefore, implies that future cardio-therapeutic interventions should seek to either acutely target the preservation of the bulbospinal sympathetic pathways (i.e., neuroprotection, transplantation, repair, and regeneration) or chronically target the spinal sympathetic pathways (i.e., neuromodulation) to offset reductions in cardiac function post SCI.

Collectively, our experiments provide important advancements to understanding the cardiac phenotype that develops post SCI. The immediate reduction in bulbospinal sympathetic control caused by the SCI itself reduces LV pressure-generating capacity and LV contractility and initiates the cellular cascades to upregulate cardiomyocyte protein degradation. Over time, the impaired control of bulbospinal sympathetic pathways likely combines with reductions in plasma volume and/or changes in circulating hormones to reduce LV chamber size and cardiomyocyte size, which further compromises the available cardiac reserve and may explain the increased disposition that SCI individuals have for cardiac arrhythmias[59] and cardiovascular disease[22]. The decrease in cardiac reserve has important clinical implications as it can limit the ability to perform regular activities of daily living and consequentially compromise the body's responses to physiological stressors[60]. By defining the temporal changes in cardiac structure and function across species and ascertaining the role the SNS plays in post SCI cardiac function, we have identified that future interventions should target the loss of sympathetic control with a view to offsetting the changes that occur in the heart.

## Methods

**Ethical approval.** Clinical protocols were approved by the University of British Columbia Clinical Research Ethics Board (H13-03072 and H13-02991) and conducted in accordance with the second Helsinki Declaration[61]. Individuals provided written informed consent prior to data collection. Preclinical protocols were approved by the Animal Care Committee of the University of British Columbia (A18-0344 and A14-0152) and conducted in strict accordance with the Canadian Council on Animal Care.

## Experimental designs

*Temporal progression of cardiac consequences post SCI.* The clinical aspect of the time-course study was conducted on human participants 18–60 years of age who had sustained a traumatic motor-complete (American Spinal Injury Association Impairment Scale [AIS] A/B) SCI between C3-C8, as classified using the International Standards for Neurological Classification of SCI[62]. Participant characteristics are reported in Supplementary Table 1. Individuals in the sub-acute group (<5 months TSI; $n = 23$; $n = 14$ male, age range = 28–60 yrs) were recruited from the GF Strong Rehabilitation Centre, Vancouver, BC, Canada from April 2014-December 2018. Individuals in the chronic group (≥24 months TSI; $n = 22$; $n = 15$ male, age range = 22–58 yrs) were recruited from the community via the

Blusson Spinal Cord Centre (BSCC), Vancouver, BC, Canada from April 2014 to September 2016. The non-injured controls ($n = 14$; $n = 11$ male, age range = 22–63 yrs) were recruited from the community via the BSCC from February 2017 to April 2018 and were both age and sex matched to SCI individuals. Exclusion criteria comprised any history of cardiovascular disease, which was confirmed with a verbal medical history, and any language or cognitive barrier that prevented the individual from following English instructions. Participants with SCI were compensated with a monetary honorarium for dedicating their time to the study. Non-injured controls participated without any financial compensation. Using a cross-sectional design, we assessed cardiac volumes, function, and mechanics in individuals with sub-acute or chronic SCI, and in non-injured controls via transthoracic echocardiography (Fig. 2a).

The preclinical aspect of the time-course study was conducted on male Wistar rats (age = 10–11 weeks, mass = 300–400 g; Envigo, USA), which were randomly assigned to undergo either a dorsal durotomy with no SCI (SHAM) or T3 complete transection SCI (T3-SCI). To investigate the temporal changes that occur in the heart post SCI, T3-SCI rats were terminated at different timepoints along the acute-to-chronic continuum: at 12 h ($n = 9$), 1 day ($n = 8$), 3 days ($n = 10$), 5 days ($n = 7$), 7 days ($n = 9$), and 8 weeks post SCI ($n = 10$). SHAM rats were terminated at 7 days ($n = 6$) and 8 weeks post-injury ($n = 7$) where appropriate for age-matching. Cardiac volumes were longitudinally assessed via in vivo echocardiography in the 7 day SHAM and T3-SCI rats at pre-injury, 1, 2, 4, and 6 days post-injury, and in the chronic 8 weeks SHAM and T3-SCI rats at pre-injury and 8 weeks post-injury (Fig. 3a). At termination, the 1 day, 3 days, 5 days, 7 days, and 8 weeks T3-SCI rats, and 7 days SHAM rats were assessed for peripheral hemodynamics and LV function via invasive in vivo catheterization (Fig. 4a). Following termination in all groups, blood was collected via cardiac puncture for circulatory plasma NE analysis via ELISA (Fig. 4d). Next, the heart was collected, and assessed for gene expression of protein degradation via qPCR and cardiomyocyte structure via immunohistochemistry (Fig. 5a).

*Bulbospinal sympathetic control and LV function in SCI.* We conducted three rat studies in which we assessed hemodynamics and LV function via invasive in vivo LV catheterization. First, we compared rats pre- and post complete transection SCI at the T3 level as well as following a subsequent I.V. infusion of the ganglionic blocker HEX. Second, we compared rats with a complete transection SCI at the T3 vs. L2 spinal level (reduced vs. intact cardiovascular SNS control, respectively). Third, we compared rats with T3 severe midline contusion SCI treated with the neuroprotective drug minocycline or a vehicle. In humans, we assessed beat-by-beat hemodynamics via finometry and ECG to estimate LV function via model flow before and during *PVS* in ten individuals with chronic cervical SCI (Fig. 9).

The HEX study was conducted in a mix of Sprague Dawley ($n = 4$; age = 23–32 weeks, mass = 440–475 g; Envigo, USA) and Wistar rats ($n = 3$; age = 11–12 weeks, mass = 350–450 g; Charles River, Canada) which first underwent in vivo LV catheterization followed (to enable pre-intervention measures) by a T3-SCI and then an infusion of HEX (I.V. 20 mg/kg) (Fig. 6a, b). LV functional measures were assessed at pre-, post SCI, and post SCI + HEX.

The level of injury study was conducted in Male Wistar rats (age = 10–11 weeks, mass = 300–400 g; Envigo, USA) which underwent either a complete spinal cord transection at the T3 (interrupted cardiac SNS control; $n = 6$) or L2 level (intact cardiovascular SNS control; $n = 7$), or SHAM injury ($n = 7$) (Fig. 7a). At 12 weeks post-injury (termination), in vivo LV catheterization was performed to assess peripheral hemodynamics and LV function.

The minocycline study was conducted in Male Wistar rats (age = 10–12 weeks, mass = 250–350 g; Charles River, Canada) which underwent T3 severe midline contusion SCI (400 kdyn, 5 s dwell) (Fig. 8a). Contused rats were randomly assigned to either a vehicle-treated ($n = 6$) or minocycline-treated group ($n = 5$). Treatment was injected I.P. every 12 h for 2 weeks and started at 1 h post SCI. Minocycline dosage was 90 mg/kg upon first injection and 45 mg/kg dose every 12 h for 2 weeks (diluted in distilled water, 30 mg/mL). At 8 weeks post-contusion (termination), in vivo catheterization was performed to assess peripheral hemodynamics and LV function. A subset of animals ($n = 4$/group) underwent retrograde tracing with FluoroGold at day 53 post-contusion. Following termination, the brainstem was collected for FluoroGold histological analysis of RVLM (located with cresyl violet stain) sympatho-excitatory neurons with additional dopamine beta-hydroxylase (DBH) and neuronal nuclei (NeuN) stains.

For the PVS study, we conducted a secondary analysis of a previous prospective study. The study included 10 male individuals with chronic cervical SCI (AIS A, $n = 5$; AIS B, $n = 5$; mean time since SCI = 15 ± 10 years [range: 4–31 years]) (Fig. 9). Cardiovascular data was collected continuously pre-PVS and during PVS.

## Animal care and spinal cord surgeries

*Care.* Animals were received at the facility at least 7 days prior to surgery to allow time for acclimatization and were group-housed in temperature-controlled rooms with 12-h light-dark cycles and social/physical enrichments. Water and food were provided ad libitum throughout all studies. Starting 3 days prior to the SCI/SHAM surgery, animals were administered daily prophylactic enrofloxacin (Baytril; 10 mg/kg, S.C., Associated Veterinary Purchasing, Langley, BC, Canada). Animal care was conducted in accordance with the Animal Care Committee of the

University of British Columbia and the standard procedures for care of high-level SCI rats, developed at our animal care facility[63].

*Complete transection SCI surgeries.* Rats were anesthetized with inhalant isoflurane (4–5% with 2 L/min $O_2$ in the induction chamber, followed by 1–2% for maintenance via a nose cone), prepped for surgery and administered buprenorphine (Vetergesic; 0.02 mg/kg, S.C., Ceva Animal Health Inc., Cambridge, ON, Canada). A detailed protocol for *T3* complete transection SCI surgery has been published by our group previously[7]. Briefly, a dorsal durotomy was performed via an incision at the dorsal midline from C8 to T2 vertebrae or from L1 to L2 vertebrae, and the dura was snipped between the T2 and T3 spinal levels or between the L1 and L2 spinal levels. Next, the animal underwent a complete transection of the T3 or L2 segment using micro scissors, and a vacuum-powered suction was used to remove ~0.5 cm of tissue. Using a dissecting microscope, complete transection was confirmed by ensuring no tissue remained between the rostral and caudal ends of the spinal cord. Hemostatic Gelfoam (Pharmacia & Upjohn Company, Pfizer Inc., New York, NY, USA) was placed in the space between the rostral and caudal ends. SHAM animals underwent a dorsal durotomy but no SCI.

*Severe midline T3 contusion SCI and FluoroGold surgeries (minocycline study only).* We have previously published a detailed protocol[30]. Briefly, a dorsal durotomy from C8 to T2 was performed. Next, the animal underwent a severe midline T3 contusion of 400 kdyn of force with a 5-s dwell time using an Infinite Horizon impactor (Precision Systems and Instrumentation, LLC, Fairfax Station, VA24) with a 2.5 mm tip. A subset of animals underwent retrograde tracing with FluoroGold (Fluorochromes Inc., Denver, CO) 53 days post-contusion as previously described elsewhere[30]. For FluoroGold, rats underwent a dorsal durotomy via an incision at the dorsal midline of the T7 segment followed by a T8 transection. Hemostatic Gelfoam was soaked in 10 µL of 4% FluoroGold in sterile saline and placed in the space between the rostral and caudal ends of the spinal cord. After verification that the Gelfoam was in contact with the rostral end[64,65], muscle and skin were sutured. The abovementioned pre-surgery care was followed. RVLM tissue was collected at termination ~1 week post-FluoroGold which is deemed sufficient time for detecting retrograde transport in the brainstem[64,65].

*Suturing and post-care.* Suturing of the muscle and skin was performed respectively with continuous absorbable Monocryl and interrupted non-absorbable Prolene (both 4-0; Ethicon Inc., USA; removed when appropriate). Following surgery, animals were closely monitored and recovered in a thermo-regulated incubator (37 °C) until anesthesia had worn off. For 3 days post SCI/SHAM (or until termination day, whichever was sooner), animals received enrofloxacin (10 mg/kg, S.C.) once daily and buprenorphine (0.02 mg/kg, S.C.) every 12 h. Animals were monitored four times daily for pain assessment and management, behavioral and physical checks, and for SCI rats specifically, manual bladder expression until return of micturition[63] (~7–10 days).

**In vivo echocardiography**. For the time-course clinical study, an experienced sonographer performed the echocardiogram examinations using a Vivid 7 / i ultrasound unit (GE Healthcare, Mississauga, ON). Individuals enrolled in the study were transferred to an echocardiography table and rested in the left lateral decubitus position for 5 min. HR was recorded simultaneously with echocardiographic images using a three-lead ECG. Three consecutive cardiac cycles were recorded at the end of a tidal expiration, and the mean value was recorded for each parameter. All measures were performed according to American Society of Echocardiography (ASE) guidelines[66] and analyzed on EchoPAC (version 202; GE Healthcare, Horten, Norway). Indices of LV mechanics were derived from apical four-chamber and parasternal short-axis images at the level of the mitral valve (basal), papillary muscle (mid), and apex (apical). Images were analyzed using 2D speckle-tracking software in accordance with current guidelines[67]. To minimize errors in endocardial border tracing, LV volumes from participants with poorly visualized endocardial definition in the apical two-chamber view used a single plane method of measurement. If volumetric planes were not well visualized, participants were not included for those specific indices. To control for differences in HR, raw speckle-tracking traces were imported into customized post-processing software (2D Strain Analysis Tool, Stuttgart, Germany), which interpolates the data into 600 points in systole and 600 points in diastole using a standard cubic spline algorithm.

For the time-course rat study, echocardiography was performed pre-injury and acutely over the first 6 days post-injury in T3-SCI and SHAM rats (Fig. 3a). We separately imaged different T3-SCI and SHAM rats pre-injury and chronically 8 weeks post-injury (Fig. 3a). Our procedures are described in detail by our group elsewhere[68]. Briefly, rats were anesthetized with inhalant isoflurane (4–5% with 2 L/min $O_2$ in the induction chamber, followed by 1–2% for maintenance via a nose cone) and placed securely in the supine position on a temperature-regulated ultrasound platform (VisualSonics, Toronto, ON, Canada) that was integrated into a high-frequency animal ultrasound machine (Vevo 3100; FUJIFILM VisualSonics, Toronto, ON, Canada). A high-frequency cardiac transducer (MS250, 13–24 MHz; VisualSonics, Toronto, ON, Canada) was used to assess LV volumetric, functional, and structural measures from B-mode parasternal long-axis, M-mode short-axis and apical PW Doppler images. HR and respiration rate were monitored

throughout the procedure. A minimum of three beats during expiration were analyzed per rat. Images were analyzed in a blinded fashion using Vevo LAB software (VisualSonics, Toronto, ON, Canada).

**In vivo LV catheterization**. In all rat studies, animals were anesthetized with urethane (1200–1500 mg/kg, I.P., Sigma-Aldrich Corporation, St. Louis, MO, USA). Once a surgical plane was confirmed, the animal was placed in the supine position for surgery. First, tracheotomy and intubation were performed to prepare for potential supportive respiratory care (Figs. 4b-1). Second, a pre-soaked PV catheter (1.9F rat PV catheter with an ADV500 PV system, Transonic®, Ithaca, NY, USA; or, SPR-869; Millar, Inc., Houston, TX, USA) was inserted in the right carotid artery via a closed-chest approach following standard procedures[69] (Figs. 4b-2). The catheter remained in the artery until pressure stabilization was achieved for a minimum of 5 min to allow basal arterial data recording (i.e., blood pressures and HR). Next, the catheter was advanced into the LV under echocardiographic guidance. Once an optimum position in line with the long axis of the LV was identified, we performed a midline laparotomy to visualize the IVC (Figs. 4b-3). Saline-soaked gauze was then placed in the abdominal cavity to facilitate future IVC occlusions without the need for additional surgery. With these two procedures complete, we waited for 15 min to enable hemodynamic stabilization. Next, baseline load-dependent PV data were recorded for a minimum of 5 min. To assess LV load-independent function, a minimum of three IVC occlusions were performed, separated by at least 5 min, with a cotton-tip applicator. IVC occlusions cause a left and downward shift in the PV loop enabling the measurement of $E_{es}$, which is a marker of cardiac contractility. For the HEX study, LV PV data were recorded for the entire experiment and analyzed for 1 min once the traces were stabilized at pre-intervention, nadir post SCI and 30 min post SCI + HEX, once the traces were stabilized. Catheter-derived volumetric data in the HEX study was calibrated with B-mode parasternal long-axis images obtained with echocardiography taken prior to catheterization (with previously mentioned equipment). All PV data was analyzed blindly using the PVAN extension in LabChart version 8.1 (PowerLab16/35; ADInstruments, Colorado Springs, CO, USA).

**Termination of rat studies and post-mortem tissue collection**. All animals were deeply anesthetized with urethane. Exsanguination (via transcardial puncture) and a bilateral pneumothorax were then performed as secondary confirmation for euthanasia. For the time-course rat study: blood was collected via cardiac puncture into EDTA tubes (Greiner Bio-One™ VACUETTE® K3EDTA, Kremsmünster, Austria) and immediately placed on ice. Tubes were centrifuged at $699 \times g$ and 4 °C for 15 min. Plasma was aliquoted and stored at −80 °C. A piece of LV apex (unaffected by the cardiac puncture) was sectioned and submerged into RNA*later*™ solution (Thermo Fisher Scientific Baltics, Vilnius, Lithuania) at 4 °C for ≥24 h for RNA stabilization prior to storage at −80 °C for gene expression analysis. To obtain a cross-sectional cut of the heart at mid-ventricular level for histology, a transcardial perfusion was performed with phosphate-buffered saline and then with 4% paraformaldehyde for fixation. The left femur was dissected for standardization purposes of cardiomyocyte dimensions. For the minocycline study: a transcardial perfusion (same as above) was performed for fixation and collection of the RVLM to assess sympatho-excitatory neurons.

**Circulatory plasma NE**. For the time-course rat study, frozen EDTA plasma was analyzed in duplicate for NE concentrations using the BA E-5200R Norepinephrine Research ELISA (Labor Diagnostika, Nordhorn, Germany). 100 µl of plasma in a selected well volume of 100 µl, and an 18-h incubation time were used according to the protocol provided by the manufacturer. Optical density was measured with the iMark microplate reader at 450 nm as per manufacturer instruction (BioRad, CA, USA) with Microplate Manager® Software version 6.3 (MPM6, BioRad). Standard curves and concentrations were automatically calculated using MPM6 software using a four-parameter fit. Wells with optical density values that were out of the range of the standard curve were removed from analysis. Three rats out of 22 did not have duplicates. The intraplate coefficient of variation (CV) was calculated prior to any removal of duplicates and was 9.74% for standards, samples, and controls. All standards, controls, and CVs were within specification provided in the quality control report included in the kit.

**Gene expression analysis**. Gene expression analysis was performed on LV tissue from the acute T3-SCI rats (12 h to 7 days groups) from the time-course rat study to investigate the rapid changing regulation of proteolytic pathways in cardiac tissue following SCI. Total RNA extraction was performed with a piece of the LV apex following the Invitrogen TRIzol™ Reagent protocol (15596026; Life Technologies Corporation, Thermo Fisher Scientific Inc., Carlsbad, CA, USA) with the only modification being an increased incubation time of 12 h in isopropanol at −20 °C. RNA purity was assessed with the NanoDrop™ 2000 spectrophotometer (260–280 nm; Thermo Fisher Scientific Inc., Wilmington, DE, USA) prior to undergoing reverse transcription into cDNA following the SuperScript™ VILO™ MasterMix protocol (11755-50; Life Technologies Corporation, Thermo Fisher Scientific Inc., Carlsbad, CA, USA). cDNA was processed with the Applied Biosystems™ PowerUp™ SYBR® Green PCR MasterMix kit (A25780) and duplicates were quantified with the Applied Biosystems™ ViiA 7 Real-Time PCR System to

be later averaged (both: Life Technologies Corporation, Thermo Fisher Scientific Inc., Carlsbad, CA, USA). The following PCR cycling conditions were respected: 95 °C for 15 min with 40 cycles of 95 °C for 1 min, 55 °C for 30 s and 72 °C for 30 s, and target primer sequences are listed in Supplementary Table 4. Target gene expression fold changes were calculated with the ΔΔCt method relative to the SHAM group and the housekeeping gene *β-actin*.

**Cardiac immunofluorescence**. For the time-course rat study, cardiac immuno-fluorescence was performed to visualize and measure cardiomyocyte dimensions. A mid-ventricular cross-sectional disc of cardiac tissue was sent to a commercial facility for paraffin embedding (Wax-it Histology Services Inc., Vancouver, BC, Canada). Sections (7 mm thick; two per rat) underwent deparaffinization, rehydration and antigen retrieval according to the Abcam® protocol. Prior to a 2-h blocking incubation period in phosphate-buffered saline (PBS) with 1% bovine serum albumin (05479-50G; SIGMA-ALDRICH, Co., St. Louis, MO, USA) and 10% normal donkey serum (NDS) (CLAS10-1864; Cerdarlane®, Burlington, ON, Canada), sections were washed with PBS with 1% triton for permeabilization. We stained to visualize: (1) the cell membranes, using wheat germ agglutinin (WGA) (Alexa Fluor® 488 conjugate 1:2000, W11261; Life Technologies Corporation, Thermo Fisher Scientific Inc., Eugene, OR, USA); (2) the Z-lines via stained α-actinin (rabbit primary 1:400, EP2529Y, Abcam, Cambridge, MA, USA; donkey anti-rabbit secondary Alexa Fluor® 546 1:1000, 711-586-152, Jackson ImmunoResearch, West Grove, PA, USA); (3) the intercalated discs via stained connexin-43 (goat primary 1:1000, NBP1-51938, Novus Biologicals, Oakville, ON, Canada; donkey anti-goat Alexa Fluor® 647 secondary 1:1000, 705-606-147, Jackson ImmunoResearch, West Grove, PA, USA); and (4) the nuclei via stained DNA (Hoechst 33342 1:10,000, H3570, Thermo Fischer Scientific Inc., Eugene, OR, USA). Positive and negative controls were performed to confirm antibody specificity. Longitudinally (located in the myocardium) and cross-sectionally (located in the sub-epicardium) oriented LV free-wall myocytes were imaged with a confocal microscope at 20x (ZEISS Axio Observer and Yokogawa Spinning Disk; ZEISS, Oberkochen, Germany). Representatives were taken with a 63x oil-objective (Fig. 5e). Cardiomyocytes were measured by one blinded individual in Fiji ImageJ (1.52e). Cardiomyocyte length was estimated by measuring the distance between two connexin-43 stained intercalated discs. Cardiomyocyte width was estimated by measuring α-actinin stained Z-lines which spanned between WGA-stained cell membranes. CSA was estimated by delineating WGA-stained cell membranes of cross-sectionally oriented myocytes using a semi-automated method (i.e., grayscale WGA image conversion, B&W threshold, and selection with Region of Interest manager). CSA analyses were limited to cardiomyocytes with a centered nucleus and stained cytoplasmic α-actinin. A minimum of 130 lengths, 210 widths, and 98 CSAs were averaged per animal. Cardiomyocyte volume was calculated with averaged length and CSA. Femur length was measured (cm) and used for standardization of myocyte dimensions[70].

**RVLM histology**. For the minocycline study, histology was performed to visualize and count RVLM descending sympatho-excitatory neurons. The protocol has been previously described by our team[30]. Briefly, the brainstem was embedded in Cryomatrix™ embedding resin (6769006; Thermo Shandon Limited, Thermo Fisher Scientific, Runcorn, Cheshire, UK) and flash frozen. Paired cross-sections of the brainstem (30 μm thick) either underwent acresyl violet stain to locate the RVLM[63,64] (20-min immersion followed by dehydration and mounting) or immunofluorescence to visualize sympathetic neurons. Sections destined to immunofluorescence were left to thaw for 1 h at room temperature, rehydrated with PBS for 10 min, and then blocked with NDS for 30 min. We stained to visualize: 1) NeuN (guinea pig primary 1:500, ABN90P, MilliporeSigma, Burlington, MA, USA; donkey anti-guinea pig secondary Alexa Fluor® 647 1:200, 706-606-148, Jackson ImmunoResearch, West Grove, PA, USA); and 2) sympatho-excitatory neurons (DBH; mouse primary 1:400, MAB308, MilliporeSigma, Burlington, MA, USA; donkey anti-mouse secondary Alexa Fluor® 488 1:200, 715-546-151, Jackson ImmunoResearch, West Grove, PA, USA). The RVLM and surrounding tissue were imaged at 100x using a ZEISS confocal microscope (same used for cardiac immunofluorescence). FluoroGold-positive, DBH-positive, and FG/DBH-positive neurons located in the RVLM and surrounding tissue were manually counted using Fiji ImageJ (15 sections per rat).

**Finometry and electrocardiogram recordings during PVS**. For the PVS study, procedures were performed in the mornings in a temperature-controlled laboratory (22–25 °C). Participants were asked to abstain from drinking alcohol on the night prior, and to abstain from drinking caffeine or smoking on the morning of PVS. Participants rested in the supine position for 10 min prior to PVS, during which pre-PVS blood pressure and HR recordings were obtained via a beat-to-beat blood pressure monitoring device (Finometer, Finapres Medical Systems BV, Arnhem, The Netherlands) and a standard 3-lead ECG (lead II; Powerlab Model ML132), and recorded on LabChart (previously described). After the rest period, either a WAHL (WAHL model 4196, Div Swenson Canada Inc, Toronto, ON, Canada) or Ferticare Clinic vibrator (Multicept APS, Rungsted, Denmark) was administered for PVS by a medical doctor with expertise in clinical sperm retrieval techniques. Participants were monitored throughout PVS for the presence of symptoms or signs of AD. This PVS method is standard in the sperm retrieval clinic at BSCC and has been previously published[71]. Blood pressure and HR were continuously monitored and recorded throughout PVS. Cardiac indices were estimated using Modelflow estimates. Analyses were performed post-study by a blinded co-author. Peaks of all measures were analyzed for 1 min at rest prior to the start of PVS (pre) and for the duration of PVS. Autonomic dysreflexia was defined as an increase in the SBP of 20 mmHg or greater[18].

**Statistics**. Data are presented as mean ± standard deviation (SD). Statistical analyses were performed using standard functions in R Studio (version 1.3.959, R Studio Team, PBC, Boston, MA) with R (4.0.1 GUI 1.72 Catalina build 7845, R Foundation for Statistical Computing, R Core Team, Vienna, Austria) with α set to 0.05. Graphical representations were made in Prism (version 6.0e, GraphPad Software, San Diego, CA, USA) and Adobe Illustrator (version 13.1.1, Adobe Inc., 2019, San Jose, CA, USA). Outliers were removed using the ±2 SD method. Normality was tested with the Shapiro–Wilk test. Equality of variances was tested with either the Bartlett or Levene test, when appropriate.

For the clinical aspect of the time-course study, the HEX study and the level of injury study, group differences were analyzed with a one-way ANOVA with Tukey HSD post hoc or Kruskal Wallis with Mann–Whitney U post hoc, if non-parametric. Human participant characteristics were analyzed using either a one-way ANOVA and Tukey HSD post hoc test (or Kruskal Wallis with post hoc test, if non-parametric) or a Mann–Whitney U t-test for TSI (non-parametric). In addition, the categorical variables of sex and American Spinal Injury Association impairment scale were analyzed using a Fisher's exact test. For the echocardiography-derived data from the acute preclinical aspect of the time-course study, group differences were analyzed using a two-way repeated-measures ANOVA. For post hoc between-group analyses, a two-sample t-test was performed, and for post hoc within-group analyses, a one-way repeated-measures ANOVA with Bonferroni corrected pairwise dependent samples t-test or Wilcoxon signed-rank test, if non-parametric, was performed. For the echocardiography-derived data from the chronic preclinical aspect of the time-course study, the absolute value comparisons which include pre-SCI and pre-SHAM data were analyzed with a two-way repeated-measures ANOVA with a two-sample t-test for post hoc analysis, while for percent change comparisons, a two-sample t-test was performed. For PV, anatomical, circulatory plasma NE, histological, and RNA fold change data from the preclinical aspect of the time-course study, a linear mixed model regression was performed and t-test values comparing the coefficients (group mean differences) of all SCI timepoints compared to SHAM were recorded. Detection of heteroscedasticity for the linear regression was performed with the Breusch-Pagan test, and if any assumptions were violated, a logarithmic or Box–Cox transformation was performed. For the minocycline study, group differences were analyzed with a two-sample t-test. If data were non-parametric a Mann–Whitney U test was performed instead. For the PVS study, differences between pre-PVS and during PVS were analyzed with a paired t-test.

Using previously published clinical data from our group with EDV as a measure[72], we determined that a sample size of nine human participants per group was sufficient for 80% power, an α of 0.05, and an effect size of 1.45 to detect a mean difference of 27 mL and SD of 5 mL between non-injured and tetraplegic individuals (G*Power, version 3.1.9.7). Using previous $E_{es}$ (key PV outcome) from our research group[7], we determined that a sample size of four rats per group for PV data was sufficient for 80% power, an α of 0.05, and an effect size of 1 to detect a significant mean difference of 0.55 mmHg μL$^{-1}$ with a pooled SD of 0.25 mmHg μL$^{-1}$ between rats with a T3-SCI and SHAM injury (G*Power).

Of the 66 animals initially allocated to groups in the preclinical temporal study, four animals died during or immediately following the T3-SCI procedure and one animal was excluded due to poor health at termination. Although we attempted to conduct all outcome assessments in every animal, we were unable to catheterize the LV in some animals, while others had poor perfusion of tissue. As such, the specific sample sizes per group are provided in their relevant figure and table captions for each outcome.

**Reporting summary**. Further information on research design is available in the Nature Research Reporting Summary linked to this article.

## Data availability
The data supporting the findings from this study are available within the manuscript and its supplementary information. Source data are provided with this paper.

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

## Acknowledgements

The authors would also like to thank all individuals with SCI for their commitment in this study. Images created with BioRender.com were inspired by Liisa Wainman. The clinical studies were supported by a grant from the Craig H. Neilsen Foundation (H13-03072; A.V.K.), the Canadian Foundation of Innovation (CFI; A.V.K.), the BC Knowledge Development Fund (BCKDF; 35869; A.V.K.), an ICORD/PRAXIS (Formerly Rick Hansen Institute) seed grant and a Canadian Institutes of Health Research grant (TCA 118348; A.V.K.). The preclinical studies were supported by the Heart and Stroke Foundation of Canada (A18-0344 and A14-0152 C.R.W.) and an ICORD/PRAXIS seed grant. Dr. West is supported by a Michael Smith Foundation for Health Research (MSFHR) Scholar Award and a Heart and Stroke Foundation of Canada National New Investigator Award. Research in Dr. West's laboratory is supported by the CFI and the BCKDF. Dr. Krassioukov holds the Endowed Chair in Rehabilitation Medicine. Mr. Shane Balthazaar is a recipient of the Robert H.N. Ho Scholarship. Dr. Williams was a recipient of MSFHR Trainee Award (17197). Dr. Nightingale was a recipient of a 2018/2020 MSFHR/Rick Hansen Foundation Research Trainee Award (17767). Dr. Walter was a recipient of a 2017/2019 MSFHR/Rick Hansen Foundation Research Trainee Award (17110).

## Author contributions

C.R.W. contributed to the conception of all preclinical studies and their designs, interpretation of data, drafting, and final editing of the manuscript. A.V.K. contributed to the conception of all clinical studies and their designs, clinical evaluation of participants with SCI, interpretation of data, drafting, and final editing of the manuscript. M.P.M.F. contributed to (1) the preclinical temporal study design, data acquisition, analyses, and interpretation; (2) the level of injury and HEX studies data acquisition, analyses, and interpretation; (3) the minocycline study data analyses and interpretation; and (4) drafting and revision of the manuscript. S.J.T.B. contributed to the clinical temporal study design, data acquisition, analyses and interpretation, drafting and revision of the manuscript. J.W.S. contributed to the minocycline study design, data acquisition, analyses and interpretation, and revision of the manuscript. A.M.W. contributed to the clinical echocardiography mechanics data acquisition, analyses and interpretation, and revision of the manuscript. M.-S.P.-M. contributed to the level of injury and HEX study designs, data acquisition, analyses and interpretation, and revision of the manuscript. T.E.N. contributed to the conception of the clinical studies and their designs, interpretation of data, drafting and final editing of the manuscript. E.E., B.H., and M.A. contributed to preclinical data acquisition and analyses, animal care, and revision of the manuscript. J.P.L. and G.S.J. contributed to the design, acquisition, analysis and interpretation of the norepinephrine ELISA, and revision of the manuscript. K.D.C. contributed to the clinical data design and acquisition, and revision of the manuscript. T.S.M.T. aided with clinical interpretation and revision of the manuscript. M.W. contributed to the conception of the clinical studies and their designs, acquisition and interpretation of data, and revision of the manuscript. M.S.R. and D.V.H. contributed to the design, acquisition and interpretation of data for both the preclinical temporal study and the level of injury study, and revision of the manuscript.

## Competing interests

The authors declare no competing interests.
