## [Peer Review File · Nature Communications]

REVIEWER COMMENTS

Reviewer #1 (Remarks to the Author):

This paper addresses the effects of spinal cord injury on cardiovascular function in human and rat subjects at the acute and chronic stages. Acute and chronic changes in left ventricular contractile function were observed in both rats and humans. It also includes studies related to the sources of neural control of these cardiovascular effects, and shows evidence of the loss of bulbo-spinal sympathetic control of cardiovascular functions, following spinal cord injury. In several subjects with chronic spinal cord injury cardiovascular responses to penile vibration induced sexual function are reported. An impressive number of anatomical, physiological assessments are made in humans and mice. RNA extraction and polymerase chain reaction (qPCR) were applied to assess gene expression related to protein degradation pathways involved in the ubiquitin-proteasome system of the LV wall. Atrogin-1 (MAFbx) was elevated 3-fold within 12 hours post-SCI. LV contractility was significantly higher, sufficient to normalize LV function, in minocycline-treated compared to vehicle-treated rats. This effect is consistent with the experiments that have demonstrated minocycline to have a neuroprotective effect of other functions, following spinal cord injury.

Other experiments were performed using pharmacological methods (hexamethonium bromide) and comparing the effects of spinal lesions at C3L2 to identify the mechanisms that might be associated with an impairment of sympathetic and parasympathetic control.

The strength of the present manuscript is the combinations of comprehensive designs of the studies performed to quantify a rather complete number of cardiovascular functions in humans and mice in both acute and chronic states of recovery from spinal cord injury. A minor weakness of the paper is the rather thin effort to discuss the impact of the changes in functions that were observed, with respect to impose limitations in daily functions and even longevity. Also, with respect to the issue of whether some of the functional losses in cardiovascular functions are mediated by load-dependent issues of the heart and cardiovascular functions in high versus low spinal cord injured individuals and with respect to levels and styles of activity of the subject after spinal cord injury. Comparisons to the adaptations to a zero gravity environment in humans and animals would also provide a useful perspective on the present data.

Otherwise, this reviewer sees no critical weaknesses of the manuscript and the results reported could provide an important contribution.

Reviewer #2 (Remarks to the Author):

Fossey and colleagues examined the link between bulbo-spinal sympathetic nervous control and cardiac function following spinal cord injury. They performed complementary studies in clinical and experimental settings. They found that spinal cord injury caused a rapid and sustained reduction in left ventricular function which was antecedent to observed structural changes. In animal studies they demonstrated that the decline in cardiac function was mediated by alterations in bulbo-spinal sympathetic control. Further pilot experiments in humans noted that activation of the sympathetic circuitry below the level of spinal cord injury caused an acute increase in systolic function. The authors concluded that their results highlight the importance of developing and implementing early interventions targeting sympathetic activation to mitigate the cardiac functional decline following spinal cord injury. The experiments appear to have been well planned and well conducted and the manuscript is well written, and the results presented in a clear and logical manner. I have a number of comments that the authors may wish to consider.

In the Introduction (lines 56-61) the authors note a number of possible physiological processes that may underpin acute cardiac events and chronic cardiovascular disease in patients with spinal cord injury. Given studies in other clinical contexts, such as heart failure, myocardial infarction or ventricular arrhythmias, where the perturbations are mediated by sympathetic excitation, it may be worthwhile including sympathetic dysreflexia as a possible factor involved in cardiac disease development. Work by Karlsson et al, combining measures of noradrenaline spillover with HRV/baroreflex function and muscle sympathetic activity pre and post interventions (eg. bladder percussion, glucose load ...) may be of relevance.

In Figure 2 it is noted that n=59 participants with 11-22/group but panels c-e appear to have less than 10. Were there factors that influenced the ability to obtain certain measures?

Figures 3 and 4, the authors document the temporal changes in cardiac function and structure. These observations are consistent with cardiac sympathetic activity and subsequent noradrenaline release impacting on the development of left ventricular hypertrophy. Do the authors have the capacity to measure markers of cardiac sympathetic activation, for instance nerve activity or plasma or tissue noradrenaline levels?

Line 158 Part II: Impaired bulbo-spinal sympathetic control is causally involved in LV functional decline post-SCI. The authors have concentrated on the RVLM as the source of sympathetic outflow. Some studies have also pointed to a role of the paraventricular nucleus of the hypothalamus as being involved in cardiac regulation. While documenting the actual central source of sympathetic drive in the setting of spinal injury may not be important given that the functional deficit occurs due to disruption to sympathetic preganglionic fibres in the intermediolateral column of the spinal cord, it may be worth considering or discussing possible alternative origins of sympathetic drive.

The authors used intra venous hexamethonium to block ganglionic transmission. Such an approach would initiate a global ganglionic blockade and as such make it difficult to tease out the role of ganglia directly controlling the heart. Are the effects observed in the heart due to disruption of sympathetic transmission directly to the heart or could the cardiac effects have occurred secondary to reflex effects of other outflows? Would injecting hexamethonium directly into the cardiac controlling ganglia be a more appropriate approach?

Figure 6 c, d shows data obtained from rats given hexamethonium 2 hours post T3-SCI. The lack of a significant difference is taken to indicate T3-SCI elicits a maximum effect on Pmax and dP/dt with no further augmentation by hexamethonium. Are the authors confident that this aspect of their study was sufficiently powered as the data at the Post 2 collection point appears to be lower than at the initial Post sampling point?

Line 204-207 – the authors note that minocycline preserved a greater number of bulbo-spinal sympathetic pathways compared to vehicle. Are these differences, shown in Fig 8 h and i, significantly different? In this study minocycline was used in a model of T3 contusion. Can these findings be extrapolated to the broader setting of spinal cord transection?

Line 217 and line 237 – the authors note that activating sublesional sympathetic circuitry improves LV function in humans with chronic SCI. This inference is made based on data showing that penile vibrostimulation (n=3) was associated with an acute rise in blood pressure, reduction in heart rate and increases in peak SV and peak dP/dt. Did the authors have data to indicate that LV function was impaired in this cohort? Is the response observed an expected physiological response to the stimulation performed? Given the significant problem associated with autonomic dysreflexia in this cohort the authors could be more circumspect in their conclusions.

Reviewer #3 (Remarks to the Author):

The authors studied the cardiac consequences of spinal cord injury (SCI) on cardiac structure and function. They included the contribution of altered bulbo-spinal sympathetic control to the decline in cardiac function. They combined experimental rat experiments and prospective clinical studies. This analysis revealed that SCI induced a significant reduction in left ventricular function before structural changes. These observations can have clinical implications.

The experimental studies are well conducted and the methodology is nicely described. The addition of clinical data is valuable. The entire presentation is coherent and clear.

My only comment is about the clinical implications: The current attitude in these patients is to give minimal adrenergic support as required to maintain adequate tissue perfusion. Should this adrenergic support be more liberal? How? With beta-adrenergic stimulation only? With which target? A normal heart rate?

Reviewer #1

Comment 1: A minor weakness of the paper is the rather thin effort to discuss the impact of the changes in functions that were observed, with respect to impose limitations in daily functions and even longevity.

Response 1: Thank you for raising these important points. Although we agree that this discussion is warranted for the field, we do not have any prospective (or retrospective) data that have truly linked changes in cardiac structure and function to hard outcomes (i.e., cardiovascular mortality) or implications for activities of daily living and thus health-related quality of life. We believe this information is missing from the wider literature and is something we are actively beginning to address in the spinal cord injury (SCI) population. Hence, we were careful not to overstate the implications of these findings. Nevertheless, we have made some subtle changes to the manuscript to better reflect this important point.

In lines 71-77 of the introduction, we now state that:

These SCI-induced alterations to cardio-autonomic function, in addition to other cardio-metabolic sequelae (e.g., alterations in physical activity¹⁰, metabolism¹¹⁻¹⁷, hemodynamics^{18,19}, and arterial stiffness²⁰) likely contribute to the increase in the incidence of acute cardiovascular events^{21,22} and the odds for chronic cardiovascular disease post-SCI²³. Such changes also limit maximal Q during exercise²⁴, which may ultimately reduce the efficacy of exercise interventions to offset cardio-metabolic disease in those with high-level SCI²⁵.

Additionally, in lines 352-354 of the discussion, we now state that:

The decrease in cardiac reserve has important clinical implications as it can limit the ability to perform regular activities of daily living and consequentially compromise the body's responses to physiological stressors⁶⁰.

Comment 2: Also, with respect to the issue of whether some of the functional losses in cardiovascular functions are mediated by load-dependent issues of the heart and cardiovascular functions in high versus low spinal cord injured individuals and with respect to levels and styles of activity of the subject after spinal cord injury.

Response 2: Thank you for your comments.

With respect to the first comment, we are not certain that we fully understand the exact request. In our preclinical experiments, in part II of the manuscript, we investigated whether the reductions in systolic function observed post-SCI were mediated by the loss of sympathetic control to the cardiovascular system as opposed to being mediated by load-dependent or hormonal changes associated with SCI. We achieved this by demonstrating that a T3-SCI, which impairs descending medullary sympathetic control of the cardiovascular system (T1-L2 innervation), leads to reduced load dependent (i.e., cardiac pressure and volumes) and load-independent cardiac function (i.e., end-systolic

elastance/contractility), whereas an L2-SCI does not. Our findings, therefore, suggest that the loss of sympathetic control mediate the reductions in cardiac load-dependent and load-independent cardiac function in high-level SCI. We provide further direct evidence of this phenomenon by demonstrating that administration of the neuroprotective agent minocycline improved cardiac systolic function in rats with high-level SCI compared to rats with the same level of injury that received vehicle treatment. Collectively, these findings suggest that reduced systolic function post-SCI is primarily mediated by loss of bulbospinal sympathetic control over the heart rather than other load-dependent changes that accompany SCI.

With respect to the second comment, for our observational clinical cohort study in part I of the manuscript, we recruited participants without a history of athletic training, thereby ensuring our findings are generalizable to the wider SCI population, who are typically reported to be physically inactive (van der Berg-Emons *et al.*, 2008, Arch Phys Med Rehabil). Furthermore, methods to capture physical activity levels in individuals with SCI display inherent limitations. For example, self-report questionnaires are influenced by recall bias and presently the accuracy of wearable devices to capture free-living physical activity in this specific population are questionable (Nightingale *et al.*, 2017, Sports Med Open). Nevertheless, we are not aware of any study that has demonstrated exercise and/or physical activity improves cardiac function in those with high-level SCI. In fact, most clinical studies demonstrate no change in cardiac function following exercise and/or physical activity interventions in those with high-lesion SCI (e.g., Williams *et al.*, 2021, Neurorehabil Neural Repair).

Comment 3: Comparisons to the adaptations to a zero gravity environment in humans and animals would also provide a useful perspective on the present data.

Response 3: Thank you for your suggestion. We agree that a zero gravity/microgravity environment is an interesting comparison to SCI, and has often been used in the literature as the ‘degree of unloading’ of the cardiovascular system is thought to be similar between microgravity and SCI. In fact, our research team has previously compared these two conditions in a review (Scott *et al.*, 2011, Spinal Cord). However, we believe these two conditions are fundamentally different phenomena that have unique implications for the cardiovascular system described below.

SCI and microgravity lead to different blood redistribution in the body. SCI causes almost immediate pooling of blood in the splanchnic region, reduced venous return and a decline in cardiac volumes (Teasell *et al.*, 2000, Arch Phys Med Rehabil), whereas spaceflight elicits an immediate cephalad redistribution of the blood volume and as such an increase in left ventricular (LV) chamber volumes (despite reduced central venous pressure) (Buckey *et al.*, 1985, J Appl Physiol). In turn, a cephalad blood redistribution alters baroreflex control of the circulation and inhibits the renin-angiotensin system (RAAS) (Watenpaugh, 1996, Oxford Univ Press). Conversely, we have previously shown that SCI causes an immediate upregulation of the RAAS system to compensate for lower blood pressure (West *et al.*, 2020, J Physiol).

Although we accept that long-duration space flight elicits cardiac remodeling and reduced blood volume (Perhonen *et al.*, 2001, *J Appl Physiol*), which is similar to that which occurs in SCI, the fundamental mechanistic causes are different. As such we do not believe this comparison provides an important contextual contribution to our findings and have therefore chosen not to include this into our discussion. Instead, we have retained our comparison to bed rest, which we believe is a more appropriate comparison to be made.

Reviewer #2

Comment 1: In the Introduction (lines 56-61) the authors note a number of possible physiological processes that may underpin acute cardiac events and chronic cardiovascular disease in patients with spinal cord injury. Given studies in other clinical contexts, such as heart failure, myocardial infarction or ventricular arrhythmias, where the perturbations are mediated by sympathetic excitation, it may be worthwhile including sympathetic dysreflexia as a possible factor involved in cardiac disease development. Work by **Karlsson et al**, combining measures of noradrenaline spillover with HRV/baroreflex function and muscle sympathetic activity pre and post interventions (eg. bladder percussion, glucose load ...) may be of relevance.

Response 1: Thank you for your suggestions. We agree that autonomic dysreflexia (AD) is associated with a significant increase in sympathetic activity and most likely contributes to negative cardiac remodeling and have in fact observed this very relationship in our own preclinical and clinical studies (West *et al.*, 2016, *Hypertension*). Although AD plays a role in the chronic setting post-SCI, we believe it is not an important factor in the acute setting as the neuro-plastic changes that occur in either peptidergic sensory afferents and/or the sympathetic pre-ganglionic neurons which are critical to express the AD reflex have not fully manifested until 14-28 days post-SCI (West *et al.*, 2015, *J Neurotrauma*; Krassioukov & Weaver, 1995, *Clin Exp Hypertens*; Krenz & Weaver, 1998, *Neuroscience*). As such, and notwithstanding the few clinical reports of AD occurring the first couple of days post-SCI (Krassioukov *et al.*, 2009, *Arch Phys Med Rehabil*), we generally believe that AD is not playing a role in altering heart function during the first weeks (rodents) or months (humans) post-SCI. It is also important to note that AD is practically the only scenario we are aware of that elicits an increase in sympathetic nerve activity post-SCI. Typically, resting sympathetic nerve activity is low post-SCI (Wallin & Stjernberg, 1984, *Brain*; Stjernberg, 1986, *Brain*; Teasell *et al.*, 2000, *Arch Phys Med Rehabil*) as are circulating catecholamines (see our new added data to the manuscript). As such, SCI is a condition of low background sympathetic tone coupled with a period of increases during episodes of AD that occur more frequently in chronic stages of SCI. As such, we do not believe sympathetic hyperreflexia is contributing to cardiac dysfunction post-SCI (it is likely that the negative effects of AD on the heart observed chronically are mediated by a different, as yet undiscovered, mechanism).

Comment 2: In Figure 2 it is noted that n=59 participants with 11-22/group but panels c-e appear to have less than 10. Were there factors that influenced the ability to obtain certain measures?

Response 2: Thank you for your comment. As is common for echocardiography imaging in individuals with SCI a number of individuals can have very challenging anatomy (i.e., calcified ribs, broken ribs, changes in lung volumes/function) that make imaging difficult. As such we were not able to obtain all views for every individual. In particular, we have reduced sample sizes in panels c to e due to sub-optimal apical 2-chamber views with poorly visible endocardial definition. We decided not to include participants for those specific indices if volumetric planes were not well visualized to reduce error in the estimation of cardiac volumes. We have clarified this in our methods and corrected the 'n' in the legend of Figure 2.

In lines 493-497 of the methods, we now state that:

To minimize errors in endocardial border tracing, LV volumes from participants with poorly visualized endocardial definition in the apical two chamber view used a single plane method of measurement. If volumetric planes were not well visualized, participants were not included for those specific indices.

Comment 3: Figures 3 and 4, the authors document the temporal changes in cardiac function and structure. These observations are consistent with cardiac sympathetic activity and subsequent noradrenaline release impacting on the development of left ventricular hypertrophy. Do the authors have the capacity to measure markers of cardiac sympathetic activation, for instance nerve activity or plasma or tissue noradrenaline levels?

Response 3: Thank you for raising this important point about measuring cardiac sympathetic activation following SCI. We have added the measurement of plasma noradrenaline for the SHAM, 1 day T3-SCI, 7 days T3-SCI and 8 weeks T3-SCI groups (Fig. 4d and Supplementary Table 5). Unfortunately, we are not in a position to be able to measure cardiac sympathetic nerve activity (SNA) due to the complexity of accessing these nerves in a closed-chest approach, which is critical to maintain for our cardiac assessments. Although splanchnic and/or renal nerve activity could be measured, we do not believe these to be useful metrics for this manuscript given the specificity with which different vascular beds regulate SNA.

The new data show that circulatory plasma noradrenaline is reduced as soon as 1 day post-SCI and remains lower chronically 8 weeks post-SCI compared to SHAM. These findings are in accordance with our *acute* cardiac functional data reported in this paper, which showed reduced LV contractility starting at 1 day post-SCI (Figure 4f), and with past studies from our research team which report reduced levels of circulatory noradrenaline in rodent models following 2-day *acute* (Hunter *et al.*, 2018, Front

Physiol) and 12-week *chronic* high-level SCI (Poormasjedi-Meibod *et al.*, 2019, J Neurotrauma).

Comment 4: Line 158 Part II: Impaired bulbo-spinal sympathetic control is causally involved in LV functional decline post-SCI. The authors have concentrated on the RVLM as the source of sympathetic outflow. Some studies have also pointed to a role of the paraventricular nucleus of the hypothalamus as being involved in cardiac regulation. While documenting the actual central source of sympathetic drive in the setting of spinal injury may not be important given that the functional deficit occurs due to disruption to sympathetic preganglionic fibres in the intermediolateral column of the spinal cord, it may be worth considering or discussing possible alternative origins of sympathetic drive.

Response 4: Thank you for your very well received comment. As we are sure the reviewer is aware, the brainstem control of cardiac function is very complex and likely involves multiple areas including the rostral ventrolateral medulla (RVLM), the paraventricular nucleus, the raphe and many others. We agree that the sole focus on the RVLM was an error in our original submission and have now amended this section to better reflect the complexity of the brainstem control over the heart. We have cautiously opted not to further discuss brainstem control over the heart post-SCI as we did not measure this in our study.

In lines 323-327 of the discussion, we now state that:

Although our current study has focused primarily on the rostral ventrolateral medulla (RVLM) as the major source of reduced sympathetic input post-SCI, there are likely several additional brainstem regions that significantly contribute to regulating sympathetic outflow and cardiovascular function post-SCI, including the paraventricular nucleus of the hypothalamus⁵⁵ and the Raphe nuclei⁵⁶.

Comment 5: The authors used intra venous hexamethonium to block ganglionic transmission. Such an approach would initiate a global ganglionic blockade and as such make it difficult to tease out the role of ganglia directly controlling the heart. Are the effects observed in the heart due to disruption of sympathetic transmission directly to the heart or could the cardiac effects have occurred secondary to reflex effects of other outflows?

Response 5: We certainly agree that our choice of hexamethonium bromide (HEX) would make it difficult to investigate the specific role of ganglia on the heart as it will block transmission from both the sympathetic and parasympathetic pre-ganglionic neurons, but leave ganglionic transmission intact. It is known from other animal studies outside of SCI, however, that the ganglia play a very small role in contributing to sympathetic nerve activity (Malpas, 1998, Prog Neurobiol) and therefore cardiac contractility. **Note**, we are not discussing the ganglia control of heart rate at the nodal tissue, which is considerably more important. We have no evidence to suggest that an SCI will alter ganglionic control of cardiac contractility and as such do not believe this is

a significant contributing factor here. It is possible, however, that the parasympathetic nervous system (PNS) applies a slight 'brake' like effect on cardiac contractility and by blocking this with HEX then it may in fact 'increase' contractility. However, the relative contribution the PNS makes to controlling myocardial contractility is likely to be small in comparison to the sympathetic nervous system (SNS) (Machhada *et al.*, 2016, J Physiol). Indeed, this phenomenon is confirmed in our present data (now including more animals, $n = 7$) where we show that the addition of HEX post-SCI does not significantly alter either maximum LV pressure or maximal rate of LV pressure generation (dP/dt_{max}), despite the HEX now blocking PNS transmission that was left intact post-SCI. Given these abovementioned points we are confident in our initial conclusion that the effects observed post-SCI are due to disruption of sympathetic transmission.

Comment 6: Would injecting hexamethonium directly into the cardiac controlling ganglia be a more appropriate approach?

Response 6: Thank you for your question. Yes, we agree that your suggested method would be a more specific approach but are unsure how this would be performed practically given the complexity involved in identifying, exposing and injecting **all** of the ganglia that control cardiac function. Indeed, without delivering a retrograde tracer/virus to the pericardium to first identify **all** of the specific ganglia involved in cardiac transmission then we deem that it would be too risky to choose one specific ganglion to target as it may leave other ganglionic control still intact, leading to an incomplete blockade. Identifying these ganglia, however, are of interest to us and something we are hoping to actively follow up on in future studies. As a first step, however, we believe our present approach to be sufficient for our experimental question.

Comment 7: Figure 6 c, d shows data obtained from rats given hexamethonium 2 hours post T3-SCI. The lack of a significant difference is taken to indicate T3-SCI elicits a maximum effect on Pmax and dP/dt with no further augmentation by hexamethonium. Are the authors confident that this aspect of their study was sufficiently powered as the data at the Post 2 collection point appears to be lower than at the initial Post sampling point?

Response 7: Thank you for raising this question. We agree that the data for the post-2 collection (HEX following a T3-SCI) had a low sample size. To remedy this concern, we have performed LV catheterization, subsequent T3-SCI and HEX infusion all in three new additional rats (longitudinal data collection). The data now include pre-intervention, post-SCI and post-HEX for a total of $n = 7$ (see new Figure 6 and Supplementary Table 8). The findings have been added to the manuscript and have been discussed in our responses above. Ultimately adding these extra data have not changed any of our prior conclusions. Note, we have decided to remove the data from the experiment which delivered HEX infusion with no previous SCI from the manuscript to reduce redundancy and confusion and as we believe the new data more clearly answer our experimental question.

Comment 8: Line 204-207 – the authors note that minocycline preserved a greater number of bulbo-spinal sympathetic pathways compared to vehicle. Are these differences, shown in Fig 8 h and i, significantly different?

Response 8: Thank you for your question. For *our minocycline study*, the histological RVLM data were only collected in $n = 4$ which we considered to be too low for proper statistical analysis as a post-hoc G*Power analysis (3.1.9.7) determined 38% power for this data (effect size $d = 1.44$). Our research team has published elsewhere, in a manuscript investigating the effects of minocycline on blood pressure control, that the drug does indeed statistically preserve 10-15% of descending sympathetic pathways (RVLM and spinal cord histology) (Squair *et al.*, 2018, J Neurotrauma). We have now added clarity of this point to the manuscript (lines 220-222):

Retrograde tracing of the bulbospinal sympathetic axons via FluoroGold injections at the T8 spinal level confirmed our previous observation that minocycline spares a greater number of descending bulbospinal sympathetic pathways compared to vehicle (Fig. 8h, i)³⁰.

Comment 9: In this study minocycline was used in a model of T3 contusion. Can these findings be extrapolated to the broader setting of spinal cord transection?

Response 9: This is a great question. We do not have a definite answer for whether minocycline would have similar effects on cardiac function following spinal cord transection compared to our findings with a severe contusion as it has not yet been investigated. The beneficial effects of minocycline have been investigated in contusion (Lee *et al.*, 2003, J Neurotrauma; Rice *et al.*, 2017, Neuroimmunol Neuroinflammation; Teng *et al.*, 2004, Proc Natl Acad Sci U S A) or compression (Wells *et al.*, 2003, Brain) SCI and spinal cord ischemia models (Takeda *et al.*, 2011, Spine [Philadelphia, Pa. 1976]), all of which are models that do not completely sever bulbospinal tracts. The increased sparing with minocycline is thought to occur by reducing secondary injury thanks to the drug's anti-inflammatory (Lee *et al.*, 2003, J Neurotrauma) and neuroprotective properties (Plane *et al.*, 2010, Arch Neurol). As a complete spinal cord transection severs all bulbospinal pathways we cautiously suspect that it will not work as effectively in this type of injury paradigm.

Comment 10: Line 217 and line 237 – the authors note that activating sublesional sympathetic circuitry improves LV function in humans with chronic SCI. This inference is made based on data showing that penile vibrostimulation ($n=3$) was associated with an acute rise in blood pressure, reduction in heart rate and increases in peak SV and peak dP/dt. Did the authors have data to indicate that LV function was impaired in this cohort?

Response 10: Thank you for your question. Unfortunately, we have not collected echocardiography data in these SCI individuals at the time of penile vibrostimulation (PVS). Our primary intent with these data were to demonstrate that when the system is activated it increases LV function (as opposed to normalizing it). We agree that making these conclusions based on an $n = 3$ with indirect methods is not ideal and as such we

have increased our sample size to 10 individuals with cervical SCI, all of which demonstrate the same pattern during PVS. We do not have non-injured PVS data as PVS is only used in individuals with disrupted neural control of the genitals and therefore not conducted in non-injured individuals. Although we do not have a control group, the obtained values for cardiovascular function in these individuals are towards the lower end of the spectrum of what would be expected in non-injured individuals (Claydon & Krassioukov, 2006, J Neurotrauma).

Comment 11: Is the response observed an expected physiological response to the stimulation performed? Given the significant problem associated with autonomic dysreflexia in this cohort the authors could be more circumspect in their conclusions.

Response 11: Thank you for your comment. While the individuals in our study had a cardio-excitatory response to PVS, we are not suggesting this be a strategy to improve cardiac function in individuals with SCI - as even in a controlled environment, these individuals are at risk for complications resulting from AD. PVS data were included in this manuscript for the sole purpose of demonstrating that activation of the sublesional sympathetic circuitry leads to increases in cardiovascular indices. This provides a direction for the focus of future research as it directly suggests the sub-lesional sympathetic circuitry is a potential therapeutic target. As such, neuro-therapeutic interventions that target these pathways may offer significant promise for alleviating cardiovascular dysfunction, as has recently been shown in a manuscript published in Nature, where epidural stimulation of the lower-thoracic spinal cord post-SCI was demonstrated to activate sympathetic preganglionic neurons and normalize blood pressure regulation (Squair *et al.*, 2021, Nature). To ensure no miscommunication of this important distinction we have clarified the following points in the manuscript in lines 331-336:

Whilst the individuals in our study had a cardio-excitatory response to PVS, the initiation of AD can be life-threatening and should therefore only be performed under carefully controlled clinical settings. Nonetheless, our PVS findings do provide compelling support for future studies to focus on activating sublesional sympathetic circuitry in a controlled fashion as a way to offset reductions in cardiac function.

Reviewer #3

Comment 1: My only comment is about the clinical implications: The current attitude in these patients is to give minimal adrenergic support as required to maintain adequate tissue perfusion. Should this adrenergic support be more liberal? How? With beta-adrenergic stimulation only? With which target? A normal heart rate?

Response 1: Thank you for raising these important questions. Current clinical guidelines for acute SCI require hemodynamic management to offset hypotension and reduced tissue perfusion via infusion of vasopressors (i.e., NE) to reach a target mean arterial pressure (MAP) between 85-90 mmHg (Resnick, 2013, Neurosurgery). Although

increasing adrenergic support is undoubtedly beneficial to cardiovascular function, NE infusions lead to increased hemorrhage in the spinal cord. We recently conducted a large multi-year study that demonstrated the acute administration of dobutamine (beta-agonistic inotropic agent), but not NE, leads to improved spinal cord oxygenation and reduced hemorrhage acutely post-SCI while normalizing cardiac contractility and chronic cardiac function post-SCI (Williams *et al.*, 2020, Nat Commun). These findings emphasize that choice of vasopressor and its MAP target are critical, and suggest that clinical studies should consider the use of beta-adrenergic stimulation for appropriate MAP management. Although we clearly find this topic very interesting we believe an extensive discussion of this topic is outside the scope of the present manuscript. We have, however, added a small mention of this in the discussion (lines 315-320) and refer readers to our prior paper which comprehensively addresses this point:

These observations extend our group's findings that beta-adrenergic stimulation improves LV contractile function post-SCI⁸, and compliment reports from studies examining heart-transplant patients (i.e., improved cardiac response to exercise post-sympathetic reinnervation⁴⁹) and sympathectomy (i.e., reduced LV contractile function post-chemical sympathectomy in rats^{46,50,51}), by highlighting the importance that the SNS plays in mediating LV contractile function post-SCI.

REVIEWERS' COMMENTS

Reviewer #1 (Remarks to the Author):

The revisions made in this version of the manuscript represents a thorough response. It combines experimental rat models as well as prospective clinical studies. The data clearly demonstrate significant and prolonged reduction in left ventricular contractility in rats. It further demonstrates the involvement of bulbospinal sympathetic control and in humans, sympathetic circuitry effects on systolic functions distal to the spinal lesion. The responses to all of the initial reviewer's comments are generally sufficient and most are excellent. The present manuscript provides clear, novel evidence of the mechanisms that contribute to the cardiovascular changes in rats and humans, which adds further to the value of the present data.

Reviewer #2 (Remarks to the Author):

I thoroughly enjoyed the authors' response. The authors have addressed my comments.

Reviewer #3 (Remarks to the Author):

the paper has improved